# Test-Time Learning for Large Language Models

**Jinwu Hu** [1 2 *]  **Zitian Zhang** [1 *]  **Guohao Chen** [1 2 *]  **Xutao Wen** [1]  **Chao Shuai** [3]  **Wei Luo** [4 2 *]
**Bin Xiao** [5 †]  **Yuanqing Li** [2 †]  **Mingkui Tan** [1 6 †]

## Abstract

While Large Language Models (LLMs) have exhibited remarkable emergent capabilities through extensive pre-training, they still face critical limitations in generalizing to specialized domains and handling diverse linguistic variations, known as distribution shifts. In this paper, we propose a **T**est-Time **L**earning (TTL) paradigm for **LLM**s, namely **TLM**, which dynamically adapts LLMs to target domains using only unlabeled test data during testing. Specifically, we first provide empirical evidence and theoretical insights to reveal that more accurate predictions from LLMs can be achieved by minimizing the input perplexity of the unlabeled test data. Based on this insight, we formulate the Test-Time Learning process of LLMs as input perplexity minimization, enabling self-supervised enhancement of LLM performance. Furthermore, we observe that high-perplexity samples tend to be more informative for model optimization. Accordingly, we introduce a Sample Efficient Learning Strategy that actively selects and emphasizes these high-perplexity samples for test-time updates. Lastly, to mitigate catastrophic forgetting and ensure adaptation stability, we adopt Low-Rank Adaptation (LoRA) instead of full-parameter optimization, which allows lightweight model updates while preserving more original knowledge from the model. We introduce the AdaptEval benchmark for TTL and demonstrate through experiments that TLM improves performance by at least 20% compared to original LLMs on domain knowledge adaptation.

---

[*]Equal contribution  [1]School of Software Engineering, South China University of Technology, China [2]Pazhou Laboratory, China [3]Zhejiang University, China [4]South China Agricultural University, China. [5]Chongqing University of Posts and Telecommunications, China [6]Key Laboratory of Big Data and Intelligent Robot, Ministry of Education, China. Correspondence to: Mingkui Tan <mingkuitan@scut.edu.cn>, Yuanqing Li <auyqli@scut.edu.cn>, Bin Xiao <xiaobin@cqupt.edu.cn>.

*Proceedings of the $42^{nd}$ International Conference on Machine Learning*, Vancouver, Canada. PMLR 267, 2025. Copyright 2025 by the author(s).

## 1. Introduction

Large Language Models (LLMs) such as GPT-4 (Achiam et al., 2023) and LLaMA (Dubey et al., 2024) have significantly advanced the field of natural language processing (NLP), demonstrating exceptional capabilities in both understanding and generating human-like text (Wang et al., 2025a). Such success is achieved through extensive pre-training on massive corpora, enabling them to learn rich language representations that facilitate superior performance in various NLP tasks (Hu et al., 2025a; Zhong et al., 2024).

Despite their impressive capabilities, LLMs face significant challenges when deployed in real-world environments with dynamic and diverse data distributions. These challenges stem from the inherent sensitivity of deep learning models, including LLMs, to distribution shifts between training and test data, often leading to substantial performance degradation (Akyürek et al., 2024). These distributional shifts manifest in two main ways: **1) Domain-Specific Terminology:** Encountering rare or specialized terms and structures in fields such as medicine or agriculture may limit the model's performance (Gu et al., 2021). **2) Linguistic Diversity Variations:** Variations in user intent and linguistic diversity, including dialects and slang, lead to distributional discrepancies that negatively affect the model's comprehension and response generation (Bella et al., 2024).

Recently, several attempts have been proposed to improve the performance of models in dynamic and diverse real-world environments. Most existing methods can be broadly categorized into four types, as shown in Table 1. *Fine-tuning* (Hu et al., 2022; Thirunavukarasu et al., 2023) adapts pre-trained models to specific tasks by updating their parameters with labeled data, but it is constrained by the need for extensive labeled datasets, limiting its practicality in dynamic environments. *Retrieval-Augmented Generation (RAG)* (Fan et al., 2024) improves performance without requiring labeled data updates by leveraging external knowledge retrieved during inference, but its success depends heavily on the quality of the retrieved information. *Test-Time Adaptation (TTA)* (Wang et al., 2021; Niu et al., 2022a; Chen et al., 2024b) adjusts model parameters during inference using only unlabeled test data, allowing the model to adapt to distribution shifts in real-time. However, most TTA

*Table 1.* Characteristics of problem settings for adapting trained models to potentially shifted test domains.

| Setting | Knowledge | Source Data | Target Data | Training Loss | Testing Loss | Learning Type |
|---|---|---|---|---|---|---|
| Fine-tuning | ✗ | ✗ | $x^t, y^t$ | $\mathcal{L}(x^t, y^t)$ | – | Supervised |
| Retrieval-Augmented Generation (Fan et al., 2024) | ✓ | ✗ | $x^t$ | – | – | – |
| Test-Time Adaptation (Wang et al., 2021) | ✗ | ✗ | $x^t$ | ✗ | $\mathcal{L}(x^t)$ | Unsupervised |
| Test-Time Training (Hardt & Sun, 2024) | ✓ | $x^s, y^s$ | $x^t$ | ✗ | $\mathcal{L}(x^t; x^s, y^s)$ | – |
| Test-Time Learning (Ours) | ✗ | ✗ | $x^t$ | ✗ | $\mathcal{L}(x^t)$ | Self-supervised |

methods rely on entropy minimization as the optimization objective, which overlooks the autoregressive dependencies within LLMs, limiting its effectiveness in improving performance on dynamic tasks (see Figure 1a). *Test-Time Training (TTT)* (Hardt & Sun, 2024; Hübotter et al., 2024) retrieves data relevant to the input from the training set or knowledge base during inference to fine-tune the model, improving its performance in dynamic scenarios. However, these methods assume that the model's training data or knowledge data is accessible, which is often not the case in practice, and they also incur additional retrieval overhead.

Although recent methods address distributional shifts in test data, they still face the following limitations: **1) Difficulty in acquiring labeled data**: High-quality labeled data for SFT of LLMs, especially for domain-specific tasks, is time-consuming, and becomes more difficult in online model updates. **2) Neglecting autoregressive dependencies**: Many existing methods, such as TTA, overlook the autoregressive nature of LLMs, leading to potential harm when using entropy minimization for parameter updates. **3) High training overhead and catastrophic forgetting:** Many methods require substantial computational resources to update model parameters and may suffer from catastrophic forgetting.

To address these limitations, we propose a **T**est-Time **L**earning (TTL) method for Large **L**anguage **M**odels, namely **TLM**, which dynamically adapts LLMs using only unlabeled test data. Specifically, we provide empirical evidence and theoretical insights to reveal that more accurate autoregressive predictions from LLMs can be achieved by minimizing the input perplexity of the unlabeled test data (see **Observation 1**). Based on this insight, we formulate the TTL process of LLMs as input perplexity minimization, enabling self-supervised enhancement of LLM performance. Furthermore, we observe that high-perplexity test samples contribute more significantly to model updates compared to low-perplexity samples (see **Observation 2**). Building on this observation, we propose a Sample Efficient Learning Strategy that employs a perplexity-based weighting scheme to actively select and emphasize high-perplexity test samples for backpropagation, thereby facilitating efficient parameter updates during Test-Time Learning. Moreover, we observe that during the Test-Time Learning, Low-Rank Adaptation

(LoRA) (Hu et al., 2022) is more effective at mitigating catastrophic forgetting compared to full parameter updates (see **Observation 3**). Based on this, we utilize LoRA for TTL parameter updates, enabling lightweight training and effectively mitigating catastrophic forgetting, in contrast to updating the full parameters of LLMs. Lastly, we construct a comprehensive benchmark named AdaptEval for TTL.

We summarize our main contributions as follows:

- **Empirical Insights on Input Perplexity Minimization**: We empirically demonstrate that output perplexity can be reduced by minimizing input perplexity. Based on this insight, we adopt input perplexity minimization as the optimization objective, enabling LLMs to adapt effectively to target domains during test time.

- **Sample Efficient Learning Strategy with High-Perplexity Focus**: We propose a Sample Efficient Learning Strategy using a perplexity-based weighting scheme to select high-perplexity test samples for update, ensuring efficient utilization of computational resources. Moreover, we use LoRA to enable lightweight training and mitigate catastrophic forgetting.

- **Benchmark and Experimental Validation of Test-Time Learning**: We establish the **AdaptEval** benchmark for TTL and demonstrate through experiments that our TLM improves performance by at least 20% over original LLMs on domain knowledge adaptation.

## 2. Related Work

The adaptability of deep learning models to dynamic and diverse real-world environments has emerged as a prominent focus in recent research. Various methods have been proposed to enhance model performance under distributional shifts, as summarized in Table 1.

**Fine-Tuning** adapts pre-trained models to specific tasks or domains by updating their parameters, such as LoRA, with labeled data (Hu et al., 2022; Thirunavukarasu et al., 2023; Chen et al., 2024c; Wang et al., 2025b). This approach allows models to specialize in domain-specific tasks by leveraging transfer learning to refine their capabilities.

However, it is often constrained by the need for extensive labeled datasets and high computational costs, which limit its practicality in dynamic environments where data distributions are continuously evolving. **In contrast, our work aims to dynamically update the model at test-time using unlabeled input data**, eliminating the need for extensive labeled datasets and addressing the challenges posed by evolving data distributions.

**Retrieval-Augmented Generation (RAG)** incorporates external knowledge by retrieving relevant information from a knowledge base during inference (Jiang et al., 2024; Qian et al., 2024; Asai et al., 2024). This allows the model to generate more accurate and contextually grounded responses without requiring parameter updates. Qian et al. (2024) propose MemoRAG, a retrieval-augmented generation framework enhanced by long-term memory for improved task performance. RAG is effective for tasks requiring up-to-date or domain knowledge but relies heavily on the quality of retrieved information and incurs additional computational latency, limiting its suitability for time-sensitive tasks.

**Test-Time Adaptation (TTA)** dynamically updates model parameters during inference by utilizing unlabeled test data (Wang et al., 2021; Niu et al., 2022a; 2023; Chen et al., 2024b;a; Liang et al., 2024; Yi et al., 2024). This approach enables real-time adaptation to distributional shifts, making it suitable for scenarios where labeled data is unavailable or the test data distribution deviates significantly from the training distribution. Wang et al. (2021) propose the test entropy minimization method, which improves model confidence by minimizing prediction entropy through online updates of normalization statistics and affine transformations. Most TTA methods rely on entropy minimization, but this approach is not well-suited for the dynamic updates required by LLMs, as shown in Figure 1a. To address this issue, we propose minimizing the perplexity of test samples as the optimization objective, which effectively enhances the performance of LLMs in dynamic environments.

**Test-Time Training (TTT)** retrieves relevant data from the training set or a knowledge base during inference and uses it to fine-tune the models (Niu et al., 2022b; Hardt & Sun, 2024; Hübotter et al., 2024). This allows the model to leverage adjacent data to better adapt to current test inputs, improving its performance in dynamic scenarios. Hardt & Sun (2024) propose a test-time training approach for LLMs by fine-tuning the model on retrieved nearest neighbors from a large-scale text embedding index. Hübotter et al. (2024) propose SIFT, a data selection algorithm that optimizes information gain to outperform Nearest Neighbor retrieval for test-time fine-tuning with low computational overhead. However, TTT assumes that the model's training data or knowledge base is accessible during deployment, and the retrieval process introduces computational overhead, which

is often impractical. **Unlike TTT, our focus is on TTL, which dynamically updates LLMs during test time using only unlabeled test data.**

## 3. Problem Formulation

Without loss of generality, let $P(x)$ denote the distribution of the training data $\{x_i\}_{i=1}^N$, where $x_i \sim P(x)$. The $f_{\Theta^\circ}(x)$ represent a general Large Language Model (LLM) that has been supervised fine-tuned (SFT) on labeled training data $\{(x_i, y_i)\}_{i=1}^N$, with parameters $\Theta^\circ$. During training, the model $f_{\Theta^\circ}(x)$ is optimized to generate coherent and contextually appropriate sequences by predicting the next token in an autoregressive manner, effectively fitting the training data and generalizing to test data from the same distribution $x \sim P(x)$. However, in real-world deployments, the distribution of test data may differ significantly from the training distribution due to various factors, leading to a phenomenon known as distribution shift. For general LLMs (*e.g.*, LLaMA and Qwen), two primary types of out-of-distribution (OOD) scenarios can occur during inference: **1) Vertical Domain Shift**: This occurs when test data contains domain-specific terminology, such as in medical, legal, or technical fields, which the model was not explicitly trained on, impairing its performance. **2) Distributional Shift in Non-Specific Domains**: Even without a specific vertical domain, factors like user intent variations and linguistic diversity (e.g., dialects, slang) can shift test data distribution from training data, affecting model understanding and response generation. In these cases, the generative performance of the model $f_{\Theta^\circ}(x)$ may deteriorate significantly because the model has not been explicitly trained to handle such distribution shifts, resulting in less coherent or contextually appropriate text generation on OOD test samples $x \sim Q(x)$, where $Q(x) \neq P(x)$.

**Test-Time Learning (TTL)** seeks to improve the performance of LLMs in the target domain by adjusting the model using only test data. Specifically, given a set of OOD test samples $\{x_j\}_{j=1}^M$, where $x_j \sim Q(x)$, the goal of TTL is to optimize the model parameters $\Theta$ to improve the quality and coherence of generated text for these test samples. Formally, TTL can be framed as the following optimization problem, where the objective is to minimize an unsupervised criterion defined over the test data:

$$\min_{\overline{\Theta}} \mathcal{L}(x; \Theta), \quad x \sim Q(x), \tag{1}$$

where $\overline{\Theta} \subseteq \Theta$ represents the subset of model parameters to be updated during the TTL. The TTL objective $\mathcal{L}(\cdot)$ can take various forms, such as minimizing the perplexity. The key challenge of TTL is to design efficient adaptation strategies that can utilize unlabeled test data to improve performance on OOD samples while maintaining training efficiency and effectively mitigating catastrophic forgetting.

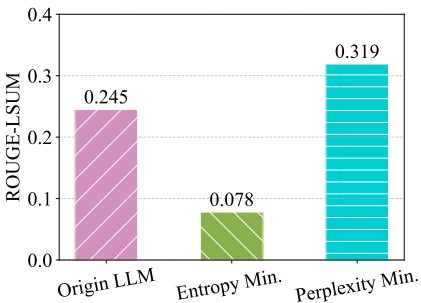

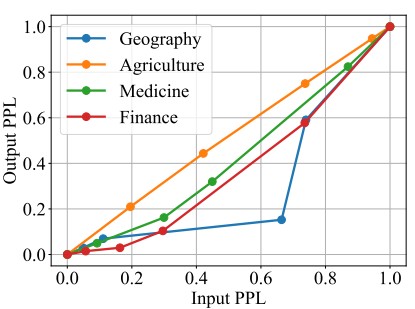

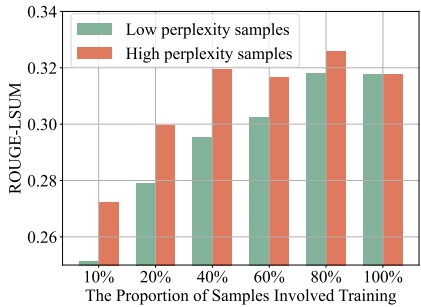

(a) Effectiveness of Entropy and Perplexity Minimization Strategies.

(b) Trends in perplexity to input and perplexity to output under Llama3.1-8B-Instruct.

(c) Effect of different test samples in test-time perplexity minimization.

*Figure 1.* Summary of our exploration and observations: (a) demonstrates that perplexity minimization improves the performance of LLMs, while entropy minimization (Wang et al., 2021) may harm their performance; (b) reveals that the trend of LLM's perplexity to the input $\mathcal{P}(x)$ and perplexity to the output $\mathcal{P}(y|x)$ is the same (results are normalized), *i.e.*, we can $\min_{\Theta} \mathcal{P}(y|x;\Theta)$ by $\min_{\Theta} \mathcal{P}(x;\Theta)$; and (c) emphasizes that training on high-perplexity samples makes more contribution than low-perplexity ones.

## 4. Test-Time Learning for LLMs

In this paper, we propose a Test-Time Learning (TTL) method for Large Language Models (LLMs) called **TLM**, which dynamically adapts LLMs using only unlabeled test data. The pipeline of TLM is shown in Algorithm 1, our proposed TLM is composed of three key components. **1)** *Input Perplexity Minimization Objective*: Inspired by the strong correlation between input perplexity and output perplexity, we adopt input perplexity minimization as the optimization objective. This enables LLMs to better fit the target data distribution during test time, as detailed in Sec. 4.1. **2)** *Sample-Efficient Learning Strategy*: Not all test samples equally impact model updates. Employing a perplexity-based weighting scheme, the model actively selects and emphasizes high-perplexity test samples for backpropagation, thereby enabling efficient parameter updates during Test-Time Learning (*c.f.* Sec. 4.2). **3)** *Lightweight Parameter Updates via LoRA*: To mitigate catastrophic forgetting and reduce computational costs, we integrate LoRA into TTL. By updating only a small subset of model parameters, LoRA enables lightweight training and effectively mitigates catastrophic forgetting, making our proposed method suitable for real-world deployment (*c.f.* Sec. 4.3).

### 4.1. Perplexity Minimization for Test-Time Learning

Perplexity (Bengio et al., 2000) is a widely used measure in language modeling that quantifies how well a model predicts a sequence of tokens (Devlin et al., 2019; Brown et al., 2020). Given a sequence of tokens $\{x_1, x_2, ..., x_T\}$, the perplexity $\mathcal{P}$ is defined as the exponentiation of the average negative log-likelihood of the predicted tokens:

$$\mathcal{P}(\{x_1, x_2, ..., x_T\}) = e^{(-\frac{1}{T}\sum_{t=1}^{T}\log p(x_t|x_{1:t-1};\Theta))}, \quad (2)$$

where $\log p(x_t|x_{1:t-1};\Theta)$ is the conditional probability of predicting token $t_i$ given the previous tokens, parameter-

---

**Algorithm 1** The pipeline of proposed TLM.

**Input:** Test samples $\mathcal{D}_{Test} = \{x_j\}_{j=1}^{M}$, the trained LLM $f_{\Theta}(\cdot)$, LoRA $\Delta\Theta$ with trainable parameters $\mathcal{B}$ and $\mathcal{A}$, batch size $B$.
1: Initialize LoRA parameters $\Delta\Theta$.
2: Add LoRA parameters to trained LLM $\tilde{\Theta} = \Theta + \Delta\Theta$.
3: **for** a batch $\mathcal{X} = \{x_b\}_{b=1}^{B}$ in $\mathcal{D}_{Test}$ **do**
4:   Calculate predictions $\tilde{y}$ for all $x \in \mathcal{X}$ via $f_{\Theta}(\cdot)$.
5:   Calculate sample selection score $S(x)$ via Eqn. (6).
6:   Update LLM ($\tilde{\Theta}$) with Eqn.(5).
7: **end for**
**Output:** The final LLM ($\tilde{\Theta}$).

---

ized by $\Theta$. A lower perplexity indicates that the model's predictions are more confident and closely align with the true distribution of the data, which implies better model fitting (Jumelet & Zuidema, 2023). Therefore, for a given question-answer pair $\{x, y\}$, minimizing the perplexity of the model's response $y$ can enhance the model's ability to fit the target data distribution, leading to improved performance on out-of-distribution (OOD) data. Specifically, by minimizing the perplexity $\mathcal{P}(y|x;\Theta)$ of the answer $y$ given the input $x$, which can be formulated as:

$$\min_{\Theta} \mathcal{P}(y|x;\Theta) = \min_{\Theta} e^{(-\frac{1}{T}\sum_{t=1}^{T}\log p(y_t|x,y_{1:t-1};\Theta))}. \quad (3)$$

This minimization process improves the model's performance in the target data distribution. *However*, during the testing phase, we can only access the user's input $x$ and not the ground truth output $y$. To address this limitation, we hypothesize that minimizing the perplexity of the input $x$, denoted as $\min_{\Theta} \mathcal{P}(x;\Theta)$, may reduce the perplexity of the model's response $y$. The mathematical justification for this transformation is based on the assumption that the model parameters $\Theta$ influence both $\mathcal{P}(y|x;\Theta)$ and $\mathcal{P}(x;\Theta)$ in a

related manner, which can be described as follows:

**Assumption 1 (Autoregressive Property):** The LLM generates each token $y_t$ based on the input $x$ and previously generated tokens $y_{1:t-1}$: $\mathcal{P}(y_t|x, y_{1:t-1}; \Theta)$. The standard next-token prediction objective makes model predictions inherently conditional on previous context quality.

**Assumption 2 (Shared Parameter Influence):** LLM parameters $\Theta$ influence both the input perplexity $\mathcal{P}(x; \Theta)$ and the conditional output perplexity $\mathcal{P}(y|x; \Theta)$. This assumption is valid across various LLM architectures, such as encoder-only and decoder-only models.

**Reducing Output Perplexity through Input Perplexity Minimization.** Minimizing the perplexity to the input $\mathcal{P}(x; \Theta)$ is equivalent to maximizing the input generation probability $P(x; \Theta)$. We employ a gradient-based theoretical analysis to formalize the intuition that question-conditioned updates benefit answer predictions, based on a key assumption. Let $\Theta' = \Theta - \eta\nabla_\Theta(-\log P(x; \Theta))$ denote the updated parameters after a single TTL step. Using a first-order Taylor expansion:

$$\log P_{\Theta'}(y|x) \approx \mathcal{O}(\eta^2) + \log P_\Theta(y|x) \tag{4}$$
$$+ \eta \underbrace{\left[\nabla_\Theta \log P(x; \Theta)\right]^\top \nabla_\Theta \log P_\Theta(y|x)}_{\text{Cross-gradient term}},$$

where $y$ is the answer to the question $x$. Our core assumption is that $\langle\nabla_x, \nabla_y\rangle = \left[\nabla_\Theta \log P(x; \theta)\right]^\top \nabla_\Theta \log P_\Theta(y|x) \geq 0$ for question-answer pairs with strong semantic alignment. Under this condition, the cross-gradient term becomes non-negative, guaranteeing: $\log P_{\Theta'}(y|x) \geq \log P_\Theta(y|x)$ for small $\eta$ (We compute the gradient inner product using 400 batches (batch size = 50) of QA pairs from the Domain-Bench on LLaMA3.1-8B. Results show 98.75% of batch-samples satisfy the non-negativity condition, with average $\langle\nabla_x, \nabla_y\rangle = +5.60$).

This form is consistent with the autoregressive property in **Assumption 1**. Naturally, based on the Shared Parameter Influence in **Assumption 2**, minimizing $\mathcal{P}(x; \Theta)$ enhances the model's overall understanding and representation of $x$. This improved representation facilitates more accurate and confident next-token predictions, which is expected to reduce $\mathcal{P}(y|x; \Theta)$. To further investigate this, we conduct a preliminary study, leading to the following observation:

**Observation 1: Trend of LLM's perplexity to the input** $\mathcal{P}(x; \Theta)$ **and perplexity to the output** $\mathcal{P}(y|x; \Theta)$ **is the same.** In the context of LLMs, it is observed that the perplexity associated with the input $\mathcal{P}(x; \Theta)$ and the perplexity of the output $\mathcal{P}(y|x; \Theta)$ exhibit similar trends. Specifically, we compute the trends of the perplexity of the input $\mathcal{P}(x; \Theta)$ and the perplexity of the output $\mathcal{P}(y|x; \Theta)$ on the four collected vertical domain datasets (see Supp. B) using Llama3.1-8b-Instruct (Dubey et al., 2024) with vary-

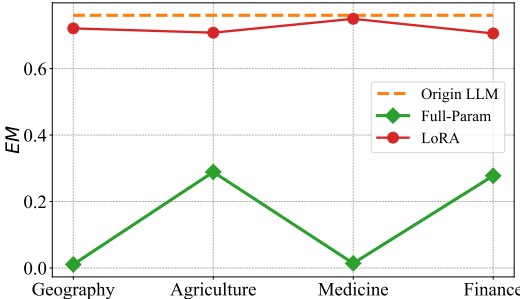

*Figure 2.* Comparison of prevent forgetting on DomainBench under Llama3.1-8B-Instruct. This observation reveals that LoRA (Hu et al., 2022) prevents catastrophic forgetting more effectively than Full-Param updates across DomainBench (see Supp. B).

ing degrees of training (for ease of presentation, we show the normalized results here). As shown in Figure 1b, the relationship between input perplexity $\mathcal{P}(x; \Theta)$ and output perplexity $\mathcal{P}(y|x; \Theta)$ demonstrates a strong positive correlation across all four vertical domains. *This indicates that reducing output perplexity is possible by minimizing input perplexity in LLMs.*

### 4.2. Sample Efficient Learning Strategy

Minimizing input perplexity $\mathcal{P}(x; \Theta)$ can enhance the performance of LLMs on target distribution data, as shown in Sec. 4.1. However, our intuition is that different test samples may produce varying effects during Test-Time Learning. To investigate this, we conduct a preliminary study, leading to the following observation:

**Observation 2: High-perplexity samples contribute more to LLM updates than low-perplexity ones.** We select different proportions of samples (the samples are pre-sorted according to their perplexity values $\mathcal{P}(x; \Theta)$) for Test-Time Learning, and the resulting model is evaluated on all test samples. As shown in Figure 1c, we find that: **1)** training the test samples with high-perplexity makes more contribution than low-perplexity ones, and **2)** training on test samples with very low-perplexity may hurt performance. The possible reason is that low-perplexity samples are already well-modeled by the pre-trained LLMs, offering little new information for further learning, which could lead to overfitting or a lack of generalization. In contrast, high-perplexity samples present more challenging data, driving greater adaptation during Test-Time Learning.

Based on **Observation 2**, we propose a Sample Efficient Learning Strategy to actively select samples for backpropagation, thereby enabling efficient Test-Time Learning. Specifically, we design an active sample selection score for each sample, denoted as $S(x)$. The criterion is that a sample should be informative for Test-Time Learning, pro-

*Table 2.* Comparison of experimental results on the DomainBench and InstructionBench of the AdaptEval (see Supp. B). We mark the better scores in bold for better visualization and easier interpretation.

| Method | DomainBench | | | | InstructionBench | | |
|---|---|---|---|---|---|---|---|
| | Geography | Agriculture | Medicine | Finance | Alpaca-GPT4 | Dolly | InstructionWild |
| Llama3.2-3B-Instruct | 0.2395 | 0.0850 | 0.1411 | 0.2229 | 0.3564 | 0.3378 | 0.2562 |
| • Tent | 0.1825 | 0.0150 | 0.1571 | 0.1093 | 0.0336 | 0.2105 | 0.0264 |
| • EATA | 0.0064 | 0.0227 | 0.0259 | 0.0149 | 0.1410 | 0.0090 | 0.0122 |
| • COME | 0.1000 | 0.1181 | 0.1542 | 0.1200 | 0.0437 | 0.2186 | 0.0697 |
| • TLM (Ours) | **0.2893** | **0.1687** | **0.2308** | **0.2953** | **0.3883** | **0.3470** | **0.2824** |
| Llama3-8B-Instruct | 0.2450 | 0.0834 | 0.1265 | 0.2329 | 0.3752 | 0.3671 | 0.2608 |
| • Tent | 0.0778 | 0.0067 | 0.0105 | 0.0372 | 0.2001 | 0.0036 | 0.0820 |
| • EATA | 0.2081 | 0.0017 | 0.0127 | 0.1257 | 0.1397 | 0.1725 | 0.1088 |
| • COME | 0.0048 | 0.0039 | 0.0301 | 0.0328 | 0.1424 | 0.0700 | 0.0240 |
| • TLM (Ours) | **0.3212** | **0.1319** | **0.2372** | **0.3242** | **0.4274** | **0.3785** | **0.2932** |
| Llama2-13B-chat | 0.2182 | 0.0840 | 0.1315 | 0.2382 | 0.3741 | 0.2892 | 0.2781 |
| • Tent | 0.0320 | 0.0196 | 0.1131 | 0.0049 | 0.0955 | 0.0076 | 0.1108 |
| • EATA | **0.2800** | 0.0771 | 0.1348 | 0.1155 | 0.0811 | 0.0513 | 0.1006 |
| • COME | 0.1981 | 0.0380 | 0.1239 | 0.0172 | 0.0806 | 0.0000 | 0.0189 |
| • TLM (Ours) | 0.2668 | **0.1013** | **0.2179** | **0.2760** | **0.3966** | **0.3007** | **0.2865** |
| Qwen2.5-7B-Instruct | 0.2649 | 0.0981 | 0.1313 | 0.2739 | 0.4439 | 0.3121 | 0.2866 |
| • Tent | 0.2362 | 0.1180 | 0.0524 | 0.1648 | 0.2132 | 0.1946 | 0.1710 |
| • EATA | 0.2109 | 0.1203 | 0.1334 | 0.2846 | 0.0000 | 0.2056 | 0.1710 |
| • COME | 0.2306 | 0.1180 | 0.0463 | 0.1780 | 0.3781 | 0.2182 | 0.1710 |
| • TLM (Ours) | **0.3081** | **0.1652** | **0.2394** | **0.3311** | **0.4608** | **0.3177** | **0.3482** |

viding enough information to drive the model's learning process, referred to as an informative sample. By setting $S(x) = 0$ for uninformative samples, we can reduce unnecessary backpropagation computations during Test-Time Learning, thereby improving the overall efficiency. Relying on the sample score $S(x)$, we use perplexity loss for model training. Then, the sample-efficient perplexity minimization is to minimize the following objective:

$$\min_{\Theta} S(x)\mathcal{P}(x; \Theta). \quad (5)$$

To obtain the active sample selection score $S(x)$, we propose a perplexity-based weighting scheme to accurately identify reliable samples and emphasize their contribution to Test-Time Learning. Formally, the active sample selection score $S(x)$ can be calculated as follows:

$$S(x) = \lambda \cdot e^{[\log \mathcal{P}(x;\Theta) - \log \mathcal{P}_0]} \cdot \mathbb{I}_{\{\mathcal{P}(x;\Theta) > \mathcal{P}_0\}}(\mathbf{x}), \quad (6)$$

where $\mathbb{I}_{\{\cdot\}}(\cdot)$ is an indicator function, $\lambda$ and $\mathcal{P}_0$ are a predefined threshold. The above weighting function excludes low-perplexity samples from Test-Time Learning and assigns higher weights to high-perplexity test samples, enabling them to contribute more significantly to model updates. It is important to note that evaluating $S(x)$ does not involve any gradient backpropagation.

### 4.3. Modulating Parameters for Test-Time Learning

**Observation3: Low-Rank Adaptation prevents catastrophic forgetting more effectively than Full-Param updates during test-time learning.** We conduct Test-Time

Learning on DomainBench (see Supp. B) using both Full-Param and Low-Rank Adaptation (LoRA) (Hu et al., 2022) updates, and evaluate the LLM's performance on GSM8K (Cobbe et al., 2021). From Figure 2, we observe that LoRA, compared to Full-Param updates, better preserves the model's originally learned general knowledge, thereby demonstrating a significant regularization effect. This is likely due to LoRA's ability to fine-tune only a small subset of model parameters, which effectively reduces the risk of overfitting and catastrophic forgetting.

Based on **Observation 3**, we adopt the LoRA for Test-Time Learning, where the optimization objective is Eqn. 5 is modified accordingly as follows:

$$\min_{\tilde{\Theta}} S(x)\mathcal{P}(x; \tilde{\Theta}) = \min_{\Delta\Theta} S(x)\mathcal{P}(x; \Theta + \Delta\Theta), \quad (7)$$

where $\Delta\Theta = \mathcal{B}\mathbf{A}$ is zero at the beginning of training, with $\mathbf{A}$ using random Gaussian initialization and $\mathcal{B}$ set to zero, and we update only $\Delta\Theta$ during the Test-Time Learning.

## 5. Experiments

### 5.1. Experimental Settings

**Datasets.** To evaluate the effectiveness of our TLM, we construct a comprehensive benchmark named **AdaptEval**, designed to cover diverse tasks and domains. AdaptEval consists of three categories of datasets. **1) DomainBench** includes four vertical domain knowledge datasets: Geography, Agriculture, Medicine, and Finance, and is designed to

Table 3. Comparison experimental results on the ReasoningBench.

| Method | ReasoningBench | | |
| | GSM8K | MetaMath | Logiqa |
|---|---|---|---|
| Llama3.2-3B-Instruct | 0.7756 | 0.7976 | 0.4194 |
| ● Tent | 0.7726 | 0.7412 | 0.4012 |
| ● EATA | 0.0032 | 0.0310 | 0.0284 |
| ● COME | 0.7710 | 0.7308 | 0.4196 |
| ● TLM (Ours) | **0.9096** | **0.8818** | **0.4572** |
| Llama3-8B-Instruct | 0.7610 | 0.6912 | 0.4550 |
| ● Tent | 0.7578 | 0.6550 | 0.4378 |
| ● EATA | 0.0250 | 0.5454 | 0.2192 |
| ● COME | 0.7479 | 0.6460 | 0.2180 |
| ● TLM (Ours) | **0.8074** | **0.7006** | **0.4868** |
| Llama2-13B-chat | 0.3458 | 0.2498 | 0.3992 |
| ● Tent | 0.2706 | 0.0040 | 0.2566 |
| ● EATA | 0.3392 | 0.0572 | 0.2606 |
| ● COME | 0.3272 | **0.2646** | 0.2462 |
| ● TLM (Ours) | **0.3508** | 0.2576 | **0.4124** |
| Qwen2.5-7B-Instruct | 0.8378 | 0.7430 | 0.5952 |
| ● Tent | 0.8455 | 0.7412 | 0.5934 |
| ● EATA | 0.7098 | 0.0070 | 0.2172 |
| ● COME | **0.8556** | 0.7559 | 0.5908 |
| ● TLM (Ours) | 0.8424 | **0.7560** | **0.6046** |

Table 4. Experimental results for the component of our proposed method on the DomainBench of the AdaptEval. The SEL means "Sample Efficient Learning Strategy" and the $(\cdot)$ indicates relative improvement over the result in the previous column.

| Version | Llama3-8B | Ours (w/o SEL) | Ours |
|---|---|---|---|
| Geography | 0.2450 | 0.3190(+30.2%) | **0.3212**(+0.7%) |
| Agriculture | 0.0834 | 0.1255(+50.5%) | **0.1319**(+5.1%) |
| Medicine | 0.1265 | 0.2326(+83.9%) | **0.2372**(+2.0%) |
| Finance | 0.2329 | 0.3222(+38.3%) | **0.3242**(+0.6%) |
| #Backwards | – | 5000 | 4772(-4.6%) |

rule, with a batch size of 1 and the learning rate of $5e{-}5/$ $5e{-}5/$ $1e{-}6$ for DomainBench/ InstructionBench/ Reasoning-Bench. The $\lambda$ and $\mathcal{P}_0$ in Eqn. 6 are set to 0.10 and $e^3$. To improve the stability of outputs produced by LLMs, we apply greedy decoding with a temperature of 0 across all experiments. More details in Supp. C.2. The source code is available at https://github.com/Fhujinwu/TLM

### 5.2. Comparison Experiments

We compare our proposed TLM, the original LLM, Tent, EATA, and COME to demonstrate the superior performance of our method. We conduct experiments on different types of datasets, including DomainBench, InstructionBench, and ReasoningBench, as summarized in Table 2 and 3. More detailed results can be found in Supp. D.

**Our proposed TLM is consistently better than the original LLMs.** From Table 2 and 3, our method consistently outperforms the original LLMs across all types of datasets and different LLM architectures. For instance, on the four datasets of DomainBench, the proposed TLM achieves at least a 20.00% improvement over the original LLMs. Specifically, on the Geography dataset, our proposed TLM improves performance by a relative 20.79% ($0.2395 \rightarrow$ 0.2893) compared to Llama3.2-3B-Instruct.

**Superior performance on Domain Knowledge Adaptation.** To evaluate the effectiveness of our proposed TLM in adapting to vertical domain knowledge, we conduct experiments on DomainBench, which includes four datasets. From Table 2, the results demonstrate that the proposed TLM outperforms the original LLMs, Tent, and EATA, achieving significant performance improvements. For example, in test-time updating of model parameters on Qwen2.5-7B-Instruct, the proposed method yields a relatively 37.32% ($0.1203 \rightarrow 0.1652$) improvement on the Agriculture dataset compared to the EATA.

**Superior performance on instruction-based task.** As shown in Table 2, our proposed TLM achieves substantial improvements over the original LLMs and Tent across all instruction-based datasets. For instance, on the Alpaca-

evaluate the LLM adaptability to specialized fields. **2) InstructionBench** contains three general-purpose instruction-following datasets: Alpaca-GPT4, Dolly, and Instruction-Wild, and focuses on the LLM adaptability to instruction-based tasks. **3) ReasoningBench** comprises three reasoning capability datasets: GSM8K, MetaMath, and Logiqa, and aims to assess the LLM logical reasoning and problem-solving abilities. These datasets collectively form a diverse and challenging evaluation suite, designed to thoroughly assess the effectiveness of TLM in adapting LLMs to tasks requiring vertical knowledge, instruction-following capabilities, and logical reasoning under distribution shifts. More details can be found in Supp. B.

**Metrics.** We use Rouge-Lsum (R-Lsum) (Lin, 2004) as the evaluation metric for DomainBench and InstructionBench, while Exact Match ($EM$) (Chang et al., 2024) is used for ReasoningBench. More metrics can be found in Supp. C.1.

**LLMs and Baseline.** We use a diverse range of LLMs of varying sizes and types, including Llama3.2-3B-Instruct, Llama3-8B-Instruct (Dubey et al., 2024), Llama2-13B-Chat (Touvron et al., 2023a), and Qwen2.5-7B-Instruct (Yang et al., 2024). We evaluate our TLM against the baseline methods, Tent (Wang et al., 2021), EATA (Niu et al., 2022a), and COME (Zhang et al., 2025). They are state-of-the-art TTA methods that update model parameters using unlabeled data. We adapt Tent, EATA, and COME to the offline setting for a fair comparison. The implementation details can be found in the Supp. C.

**Implementation Details.** We use AdamW as the update

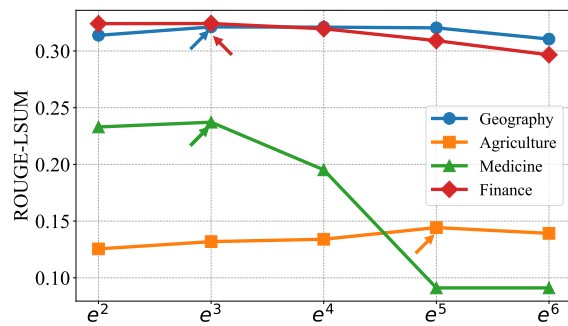

*Figure 3.* Effects of different perplexity margins $\mathcal{P}_0$ in Eqn. 6.

*Table 5.* Experimental results of our proposed method in the Online setting. Geo., Agri., Med., and Fin. represent the Geography, Agriculture, Medicine, and Finance, respectively. LLM refers to Llama3-8B-Instruct. NF4 is 4-bit NormalFloat.

| Method | DomainBench | | | | #Backwards |
| --- | --- | --- | --- | --- | --- |
| | Geo. | Agri. | Med. | Fin. | |
| LLM | 0.2450 | 0.0834 | 0.1265 | 0.2329 | – |
| Tent (Online) | 0.0804 | 0.0112 | 0.0142 | 0.0489 | 5000 |
| EATA (Online) | 0.1008 | 0.0186 | 0.0202 | 0.0815 | 4943(-1.1%) |
| Ours (Online) | **0.2787** | **0.1579** | **0.1340** | **0.2455** | **1514**(-69.7%) |
| LLM (NF4) | 0.2439 | 0.0859 | 0.1237 | 0.2325 | – |
| Ours (NF4) | **0.3069** | **0.1533** | **0.2306** | **0.3193** | **4783**(-4.3%) |

GPT4 dataset, our proposed TLM improves the performance of Llama3.2-8B-Instruct by 13.91% (0.3752 → 0.4274), showing a relative improvement of about 113.60% (0.2001 → 0.4274) compared to Tent, demonstrating its effective adaptation to general instruction-following tasks.

**Superior performance on logical reasoning task.** As shown in Table 3, our proposed TLM significantly outperforms the original LLMs and Tent on all reasoning datasets in ReasoningBench. For instance, on the GSM8K dataset, our proposed TLM improves the performance of Llama3-8B-Instruct by 6.10%, highlighting its ability to enhance logical reasoning under complex arithmetic and problem-solving tasks. These results confirm that our method not only adapts effectively to distributional shifts but also enhances the reasoning capabilities of LLMs during test time.

### 5.3. Ablation Studies

**Effectiveness of Input Perplexity Minimization.** To evaluate the effectiveness of input perplexity minimization, we conduct experiments comparing the performance of Llama3-8B-Instruct and our method without the Sample Efficient Learning Strategy (Ours w/o SEL). As shown in Table 4, input perplexity minimization significantly improves the performance of LLMs across all datasets compared to the original Llama3-8B-Instruct, demonstrating that minimizing input perplexity effectively enhances the LLM's ability to adapt to target domains. Specifically, by minimizing input perplexity during Test-Time to update the LLM parameters, we achieve a relative performance improvement of over 30% compared to the original Llama3-8B-Instruct model. Notably, on the Medicine dataset, the improvement reaches 83.9%. The effectiveness of input perplexity minimization $\mathcal{P}(x; \Theta)$ lies in its ability to enhance the LLM's understanding and representation of the input, which helps improve the model's adaptation to the target domain data.

**Effectiveness of Sample Efficient Learning Strategy in Eqn. 5.** The Sample Efficient Learning Strategy improves the efficiency of TTL by actively selecting high-perplexity samples that contribute more to the LLM's adaptation. As shown in Table 4, by prioritizing the most informative and relevant test samples for backpropagation, this strategy not only further enhances LLM performance but also reduces unnecessary computational overhead. Specifically, using the Sample Efficient Learning Strategy results in a relative performance improvement of approximately 2.0% on the target test data, while reducing the training data by about 5.0%. For instance, the relative performance improvement on the Agriculture dataset is 5.1%.

**Effects of the $\mathcal{P}_0$ in Eqn. 6.** The threshold $\mathcal{P}_0$ in Eqn. 6 plays a crucial role in controlling the threshold for sample selection during Test-Time Learning. To explore the optimal threshold for $\mathcal{P}_0$, we conduct experiments with values of $\mathcal{P}_0$ set to $\{e^2, e^3, e^4, e^5, e^6\}$. As shown in Figure 3, when $\mathcal{P}_0 = e^3$, our method achieves the best performance on three datasets, namely Geography, Medicine, and Finance, while also showing near-optimal performance on the Agriculture dataset. Therefore, we select $\mathcal{P}_0 = e^3$ for all experiments. When $\mathcal{P}_0$ is set too high or too low, it negatively affects performance. Specifically, a value that is too high restricts the number of high-perplexity samples selected, limiting the model's ability to adapt to new and complex data. On the other hand, a value that is too low includes too many low-perplexity samples, which do not contribute effectively to adaptation and could lead to inefficiencies and overfitting.

### 5.4. More Discussions

**Online Test-Time Experiments.** To further assess the performance of our TLM, we conduct experiments in the online Test-Time Learning setting. The online setting is similar to Test-Time Adaptation (Wang et al., 2021; Niu et al., 2022a), where the model processes the input to generate an output while simultaneously updating its parameters. Notably, the model parameters are updated only once every 100 test samples. From Table 5, the proposed method also achieves significant performance improvements over Llama3-8B-Instruct across different domain datasets in the online setting. Additionally, our proposed method reduces

the number of backward by 69.7% ($5000 \rightarrow 1514$) in the online setting. This is because, as the LLM is updated, some samples progressively become easier for the model, and are thus excluded from TTL in Eqn. (6)

**Experiments on Quantized LLM.** To evaluate the performance of our method on quantized LLMs, we conduct experiments on a 4-bit quantized version of Llama3-8B-Instruct, following the settings of QLoRA (Dettmers et al., 2024). From Table 5, our method also demonstrates strong performance on target domain datasets when applied to quantized LLMs. Specifically, the proposed method improves at least 25.0% cover Llama3-8B-Instruct (NF4) on four datasets on DomainBench, highlighting the broad applicability of our TTL scheme.

## 6. Conclusion

In this paper, we propose a novel Test-Time Learning (TTL) method for Large Language Models (LLMs), named **TLM**, to address the challenges posed by dynamic and diverse data distributions in real-world environments. By leveraging only unlabeled test data, **TLM** efficiently adapts LLMs and improves their robustness in target domains. Specifically, through observation and theoretical analysis, we argue that reducing output perplexity can be achieved by minimizing the input perplexity of unlabeled test data. Based on this insight, we adopt input perplexity minimization as the optimization objective for test-time LLM updates. Moreover, we find that high-perplexity test samples play a more crucial role in model updates than low-perplexity samples. This insight motivates the development of our Sample Efficient Learning Strategy, which actively selects and emphasizes high-perplexity test samples for backpropagation, optimizing the learning process. Lastly, we reveal that Low-Rank Adaptation is more effective than full parameter updates in mitigating catastrophic forgetting, and we utilize it for parameter updates during TTL, enabling lightweight model adaptation while effectively preserving prior knowledge.

## Acknowledgements

This work was partially supported by the Joint Funds of the National Natural Science Foundation of China (Grant No.U24A20327), Key-Area Research and Development Program of Guangdong Province (2018B010107001), TCL Science and Technology Innovation Fund, and the Young Scholar Project of Pazhou Lab (No.PZL2021KF0021).

## Impact Statement

This paper presents work whose goal is to advance the field of Machine Learning. There are many potential societal consequences of our work, none of which we feel must be specifically highlighted here.

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

# Supplementary Materials for
# "Test-Time Learning for Large Language Models"

In the Supplementary, we provide descriptions of more related works, details, and experimental results of the proposed TLM. We organize the supplementary into the following sections.

- In Section A, we provide descriptions of related works regarding Large Language Models, Retrieval-Augmented Generation, and Test-Time Adaptation.

- In Section B, we present a more detailed version of our AdaptEval Benchmark.

- In Section C, we present a more detailed implementation of our experiments.

- In Section D, we report more results of our experiments.

- In Section E, we provide the discussion of TLM and future directions.

## A. More Related Work

### A.1. Large Language Models

The rapid progress in natural language processing (NLP) has been marked by the emergence of large language models (LLMs), which have fundamentally transformed the landscape of artificial intelligence. These models, rooted in the Transformer architecture (Vaswani et al., 2017), leverage extensive pre-training on massive text corpora to acqire remarkable capabilities. They have shown impressive performance across a wide range of tasks (Wei et al., 2022a; Hu et al., 2025b), such as high-quality question answering (Shao et al., 2023; Peng et al., 2023), coding (Chen et al., 2021), and intermediate reasoning (Wei et al., 2022b). The unprecedented success of LLMs has spurred significant discussions regarding their application for achieving artificial general intelligence (AGI) (Zhao et al., 2023). Based on their architectural design, existing LLMs can be categorized into three major classes: encoder-only models, decoder-only models, and encoder-decoder models.

**Encoder-only models.** The encoder-only models primarily employ the Transformer encoder to encode input sequences into rich contextual representations. They are particularly effective in natural language understanding (NLU) tasks, where the focus lies in extracting semantic meaning from text. One notable example is BERT (Devlin et al., 2019), which uses bidirectional encoding to capture context from both preceding and succeeding tokens. Pre-trained on extensive datasets like BooksCorpus (Zhu et al., 2015) (800M words) and English Wikipedia (2,500M words), BERT set new benchmarks on datasets such as GLUE and MultiNLI. Subsequent iterations, including RoBERTa (Liu, 2019) and DeBERTa (He et al., 2021), introduced architectural refinements and improved pre-training strategies, further enhancing performance. Despite their strengths in understanding tasks, encoder-only models are inherently unsuited for tasks that require sequence generation, such as translation or text completion.

**Decoder-only models.** This kind of models, in contrast, rely solely on the Transformer decoder and are designed to generate text in an auto-regressive manner, where each token is generated sequentially, conditioned on previously generated tokens. Obviously These models excel in natural language generation (NLG) tasks, such as summarization, content creation and QA. The Generative Pre-trained Transformer (GPT) series (Radford, 2018; Brown et al., 2020; Achiam et al., 2023) developed by OpenAI examines this class, with GPT-3 being a landmark model that features 175 billion parameters. Trained on a diverse corpus spanning Common Crawl (Raffel et al., 2020), WebText2, Books 1, Books 2, and Wikipedia datasets. GPT-3 has demonstrated extraordinary few-shot and zero-shot learning capabilities on many language tasks. In addition to GPT series, many decoder-only models have been developed, such as OPT, LLaMA, Llama2, Llama3 from Meta (Zhang et al., 2022b; Touvron et al., 2023a;b; Dubey et al., 2024), PaLM, PaLM2 from Google (Chowdhery et al., 2023; Anil et al., 2023), BLOOM from BigScience (Le Scao et al., 2023), and Qwen series from Alibaba (Bai et al., 2023; Yang et al., 2024).

However, decoder-only models, while excelling in generative tasks, often lack the nuanced comprehension abilities required for deep understanding of long and complex input contexts.

**Encoder-decoder models.** Encoder-decoder models incorporate both the encoder and decoder components of the Transformer, combining the strengths of the two structures to handle tasks that require input understanding and sequence generation. Prominent examples of encoder-decoder models include GLM from Tsinghua University (Du et al., 2022), T5, FLAN-T5, and UL2 from Google (Raffel et al., 2020; Chung et al., 2024; Tay et al., 2023), as well as BART from Meta (Lewis et al., 2020a). For instance, GLM adopts an autoregressive blank-infilling objective to effectively address three core challenges in NLP: natural language understanding (NLU), unconditional text generation, and conditional generation. With a maximum capacity of 130 billion parameters, it is pre-trained on datasets such as BookCorpus (Tay et al., 2023) and Wikipedia. GLM surpasses BERT on the SuperGLUE benchmark by 4.6%-5.0% and demonstrates superior performance compared to FLAN-T5 on both NLU and generation tasks using fewer parameters and training data.

### A.2. Retrieval-Augmented Generation

As one of the most representative techniques in the field of generative AI, **Retrieval-Augmented Generation (RAG)** aims to enhance the quality of the generated text content with retrieved information. It achieves this by integrating two critical components: (i) a retrieval mechanism that accesses relevant documents or information from external knowledge sources, and (ii) a generative module that synthesizes this information to produce coherent and contextually accurate text (Lewis et al., 2020b). By combining these capabilities, RAG models are able to generate not only fluent and human-like text but also outputs that are grounded in up-to-date and factual data, significantly enhancing their reliability and applicability in real-world scenarios. We categorized existing RAG methods into the following two main classes according to whether training is needed for further discussion: training-free approaches and training-based approaches.

**Training-free RAG.** Training-free RAG methods address the challenges of frequent fine-tuning and updating model parameters, which require substantial computational resources and time (Lewis et al., 2020b). These approaches leverage retrieved knowledge directly at inference time by incorporating the retrieved text into the prompt, eliminating the need for additional training. As the performance of large language models (LLMs) is highly sensitive to input queries, many training-free RAG methods refine prompts by integrating external knowledge (Jiang et al., 2023; Li et al., 2023; Kim et al., 2023; Ram et al., 2023; Trivedi et al., 2023; Wang et al., 2023). For example, In-Context RALM (Ram et al., 2023) augments the generation process by prepending retrieved documents to the original prompt without altering LLM parameters. IRCoT (Trivedi et al., 2023) enhances reasoning by interleaving chain-of-thought (CoT) generation and retrieval, ensuring access to more relevant information across iterative reasoning steps. SKR (Wang et al., 2023) enables flexible utilization of both internal and external knowledge by guiding LLMs to decide whether a question can be answered based on internal knowledge before invoking the retriever. Despite their efficiency, training-free RAG methods often face limitations in optimizing the retriever and generator for specific downstream tasks, leading to suboptimal utilization of retrieved knowledge. To address this, training-based RAG approaches fine-tune both components, enabling large language models to effectively adapt and integrate external information.

**Training-based RAG.** This kind of methods aim to optimize both the retriever and generator to enhance their alignment and effectiveness. One typical success is DPR (Karpukhin et al., 2020), which employs two independent BERT (Devlin et al., 2019) encoders for queries and passages and trains them via contrastive learning. Ren et al. (2023) employs a two-stage approach, starting with the pretrain S-BERT (Reimers & Gurevych, 2019) as a retrieval backbone, enhanced by an adaptive hybrid strategy to effectively gather relevant demonstration. Next, a T5 model is used as the generator, which is further fine-tuned to align with the target labels and inputs. In contrast, RA-DIT (Lin et al., 2024) first fine-tuning LLMs to effectively use retrieved knowledge and then refining the retriever to align with the model's requirements. To address indiscriminate retrieval and the incorporation of irrelevant passages, Self-RAG (Asai et al., 2024) introduces special tokens to dynamically assess the necessity of retrieval and control its behavior. More recently, MemoRAG (Qian et al., 2024) incorporates a memory module that generates context-specific cues to link the knowledge base to precise information, improving retrieval accuracy and response quality.

Despite their advantages, RAG models rely heavily on the quality and relevance of the retrieved knowledge, as inaccuracies or irrelevant information can directly compromise the quality of the generated output. Furthermore, the dual-step process of retrieval and generation for each query introduces significant computational overhead, posing challenges for real-time and resource-constrained applications.

*Table 6.* Components of AdaptEval.

| Category | Dataset | Sources |
|---|---|---|
| DomainBench | Geosignal | Geoscience knowledge base, etc. |
| | Agriculture-QA | Agriculture data |
| | GenMedGPT-5k | ChatGPT-generated data |
| | Wealth-alpaca_lora | FiQA data, etc. |
| InstructionBench | Dolly-15k | Databricks data |
| | Alpaca_gpt4_en | GPT-4 instruction data |
| | InstructionWild | User instruction data |
| ReasoningBench | GSM8k | OpenAI data |
| | MetaMathQA | Advanced math datasets |
| | Logiqa | Civil service logic tests |

### A.3. Test-Time Adaptation

**Test-time adaptation** (TTA) aims to improve a model's performance on unseen test data, which may undergo distribution shifts, by learning directly from the test data during the testing phase. Based on their reliance on backward propagation, we categorize the related TTA works into the following two groups for discussion.

**Backpropagation (BP)-Based TTA.** A foundational approach in this category is Test-Time Training (TTT) proposed by (Sun et al., 2020). TTT involves training a source model using both supervised and self-supervised objectives during the training phase. At test time, the model is adapted using self-supervised objectives such as rotation prediction (Sun et al., 2020), contrastive learning (Liu et al., 2021; Bartler et al., 2022), or reconstruction learning (Gandelsman et al., 2022). To address scenarios where modifying the training process or accessing source data is not feasible, Fully TTA methods directly update pre-trained models during testing. These methods rely on unsupervised learning objectives, with entropy minimization (Wang et al., 2021; Niu et al., 2023) emerging as one of the most widely used techniques due to its simplicity and effectiveness. Entropy minimization encourages the model to produce confident predictions by reducing uncertainty in its output distribution. This approach effectively aligns predictions to a single class. Beyond entropy minimization, other unsupervised objectives, such as prediction consistency maximization (Zhang et al., 2022a; Fleuret et al., 2021) and feature distribution alignment (Mirza et al., 2023), have also been explored, enhancing the model's ability to adapt to diverse test-time scenarios.

To further enhance the efficiency of backpropagation-based TTA, recent research efforts have focused on tow primary aspects: (1) Sample Efficiency: As not all test samples contribute equally to adaptation. Several recent works (Niu et al., 2022a; 2023; Shu et al., 2022; Lee et al., 2024) have introduced sample selection strategies to focus on reliable and non-redundant samples, reducing the noise in the gradient and the number of samples for TTA, thereby enhancing adaptation performance and efficiency. (2) Memory Efficiency: Addressing the memory-intensive nature of backpropagation, methods such as EcoTTA (Song et al., 2023) optimize parameter-efficient components during adaptation, while MECTA (Hong et al., 2023) reduces batch size to lower memory consumption. Additionally, MECTA introduces a domain-aware batch normalization layer to stabilize model updates, even with smaller batch sizes. Similar to the sample efficiency methods, we propose a Sample-Efficient Learning Strategy that uses a perplexity-based weighting scheme to prioritize high-perplexity test samples for backpropagation, ensuring efficient utilization of computational resources.

**Backpropagation-Free TTA.** The development of BP-free TTA has seen significant progress, with early research focusing primarily on the adjustment of batch normalization (BN) layer statistics using test data. These methods primarily involved recalculating the mean and variance of BN layers based on testing data (Nado et al., 2020; Schneider et al., 2020). However, such approaches were limited to adapting BN layers, restricting their applicability to architectures that heavily rely on BN. To overcome these limitations, more generalized methodologies have been proposed to enhance the flexibility and effectiveness of BP-free TTA. For instance, reconstruction-based approaches focusing on input-level adaptation leverage advanced techniques like diffusion models to preprocess corrupted test images before prediction (Gao et al., 2023; Oh et al., 2025). Additionally, output-level adaptation methods have been developed, such as T3A (Iwasawa & Matsuo, 2021), which utilizes prototype-based classifier adjustment for adaptive predictions, and LAME (Boudiaf et al., 2022), which directly corrects the predicted logits. The recent advanced FOA (Niu et al., 2024) adapts models to unseen test samples without

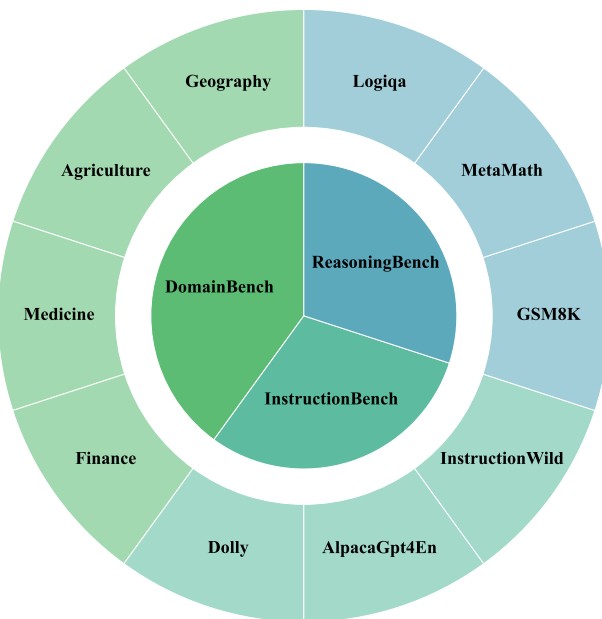

*Figure 4.* Distributions of AdaptEval.

backpropagation by learning a prompt through a derivative-free covariance matrix adaptation strategy and adjusting model activations to align with the source training domain. Looking forward, there is great potential to extend our method to the realm of efficient BP-Free TTA, thereby further broadening the practical applicability of our approach in diverse real-world scenarios.

## B. AdaptEval Benchmark

To the best of our knowledge, no existing benchmark is specifically designed to evaluate the adaptability of Large Language Models (LLMs) across diverse data distributions. Since the diversity of tasks and domains inherently captures variations in data distributions, we address this gap by introducing a comprehensive benchmark, **AdaptEval**, which spans a wide range of tasks and domains to thoroughly assess the effectiveness of our proposed **TLM**. AdaptEval is designed to capture two primary types of out-of-distribution (OOD) scenarios at test time: vertical domain shift and distributional shift in non-specific domains, as described in the previous section. To build a diverse and challenging evaluation framework, we collect high-quality datasets from HuggingFace, ensuring coverage across various data distributions. Specifically, AdaptEval consists of three categories of datasets: **DomainBench**, **InstructionBench**, and **ReasoningBench**. These categories are tailored to evaluate LLMs' adaptability to tasks requiring vertical knowledge, instruction-following capabilities, and logical reasoning under distribution shifts. A summary of the datasets included in AdaptEval is presented in Table 6, with further analysis provided below.

AdaptEval consists of the following three core categories, as shown in Figure 4.

- **DomainBench.** This category includes four vertical domain knowledge datasets: Geography, Agriculture, Medicine, and Finance. It evaluates the adaptability of LLMs to specialized fields by assessing their ability to handle tasks requiring domain-specific expertise, such as named entity recognition, judgment, and question answering. By incorporating domain-specific terminology and real-world complexities that may challenge model performance, DomainBench provides a rigorous evaluation of models' proficiency in mastering and applying specialized knowledge.

- **InstructionBench.** This category comprises three general-purpose instruction-following datasets: Alpaca-GPT4, Dolly, and InstructionWild. It evaluates the adaptability of LLms to instruction-based tasks by assessing their ability to comprehend, interpret, and execute a diverse range of user instructions. The datasets cover various task types, such as question answering, text classification, and summarization, while introducing variations in user intent, phrasing, and

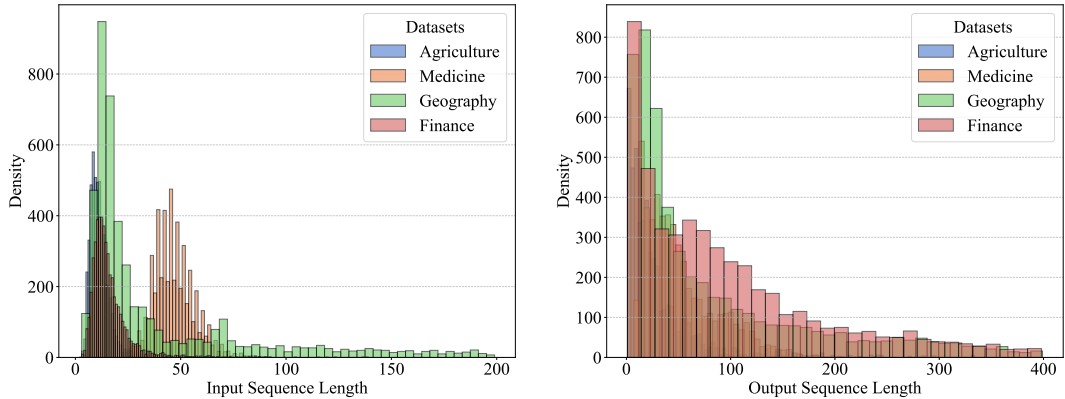

*Figure 5.* Distribution of Sequence Lengths for Samples in DomainBench.

linguistic styles, providing a thorough assessment of the model's capacity to process and respond effectively to diverse instructions in real-world scenarios.

- **ReasoningBench.** This category contains three reasoning-focused datasets: GSM8k, MetaMath, and Logiqa, designed to evaluate the logical reasoning and math problem-solving abilities of LLMs. It evaluates the model's ability to handle intricate reasoning processes through diverse scenarios, including multi-step mathematical reasoning, complex math problem solving, and logical reading comprehension. ReasoningBench evaluates tasks that require precise and consistent reasoning, offering a thorough test of a model's ability to tackle complex problems and produce logical, accurate solutions.

### B.1. DomainBench

DomainBench focuses on evaluating the model's adaptability and performance in four vertical domains: Geography, Agricultural, Medical, and Financial. To ensure the comprehensiveness and scientific rigor of the evaluation, DomainBench integrates four meticulously selected datasets: GeoSignal, Agriculture-QA, GenMedGPT-5k, and Wealth-Alpaca_Lora. Each dataset is sourced from a broad range of specialized domains, enabling the measurement of large model performance on complex domain-specific knowledge and task execution. The distribution of sequence lengths for the dataset samples and an example table of dataset entries are provided in Figure 5 and Table 7.

**Geography**: The GeoSignal[1] dataset is a knowledge-intensive instruction-tuning resource tailored for the Earth Sciences domain, aiming to improve model performance in this field. It comprises approximately 39.7k samples, created through a mix of human curation and semi-automatic methods. The dataset is designed to align with user intent, featuring two sections: a general section for human instruction alignment and a professional section focused on Earth Sciences expertise. It includes tasks such as Named Entity Recognition (NER), relation inference, fact verification, and question answering, enriched with domain-specific terminology like "volcanic neck" and "geomagnetic elements". Data sources span a multi-modal Earth Sciences knowledge graph (GAKG), academic resources (DDE Scholar), and various databases and QA platforms. From this dataset, we randomly select 5k samples to form the *Geography* dataset, which evaluates the model's domain knowledge and task performance in Geography.

**Agriculture**: The Agriculture-QA[2] dataset focuses on agricultural QA, containing about 22.6k samples. It covers various aspects of agricultural production, such as crop cultivation, livestock farming, soil management, and farming practices. These QA tasks challenge the model's ability to transfer knowledge and comprehend natural language, especially within the highly specialized agricultural field. We randomly select 5k samples to create the *Agriculture* dataset, aiming to evaluate the model's performance in handling agricultural QA tasks.

---

[1]https://huggingface.co/datasets/daven3/geosignal
[2]https://huggingface.co/datasets/KisanVaani/agriculture-qa-english-only

*Table 7.* Examples of DomainBench data.

| | |
|---|---|
| Geography | **Input:** I wonder about the fossil of the Odonata. Can you help?
**Output:** Certainly, the Odonata has a fossil of Odonata– Wolfe et al. 2016. |
| Agriculture | **Input:** Why do cassava propagated through stem cuttings mature faster compared to seed propagation?
**Output:** This is because the stem cuttings already have a head start in development and can quickly establish roots and start growing. |
| Medicine | **Instruction:** If you are a doctor, please answer the medical questions based on the patient's description.
**Input:** Doctor, I'm experiencing some strange symptoms. I've been acting hostile towards my family, and I'm having problems with my movement. Also, I've noticed that I've been slurring my words lately.
**Output:** Based on what you've described, it sounds like you may be experiencing delirium, which is a state of mental confusion that can cause changes in behavior, movement problems, and difficulty speaking clearly. We'll need to run some tests to determine the underlying cause, but it's important that we act quickly to manage your symptoms and prevent any complications. |
| Finance | **Input:** Assemble an outfit for a Halloween party.
**Output:** For a Halloween party, assemble an outfit that will make you stand out! Start with a base layer of black trousers and a white t-shirt. Then pair it with an eye-catching statement piece, such as a bright orange blazer. For shoes, choose something with a spooky vibe, like a pair of lace-up boots with gothic buckles. Accessorize with a deep red scarf and a black fedora. Finish off the look with a classic cane and a golden pocket watch. |

**Medicine**: GenMedGPT-5k[3] with a total of 5.45k samples is a medical dialogue dataset generated by ChatGPT, and is designed to emulate real-life conversations between patients and doctors. It provides rich, scenario-specific content that closely mirrors real medical scenarios. The dataset encompasses a wide range of medical queries from patients and responses or diagnoses from doctors. We randomly select 5k samples to create the *Medicine* dataset, aiming to assess the model's performance in medical dialogue, particularly its adaptability to medical contexts and the quality of its responses.

**Finance**: The Wealth-Alpaca_Lora[4] dataset is focused on the financial domain, consisting of 44.3k samples. It integrates general task data (Alpaca dataset), financial domain data (FiQA dataset), and custom task data generated using GPT-3.5. This dataset is extensively used for sentiment analysis, opinion mining, and QA tasks in financial texts, covering a variety of real-world applications in finance. We randomly select 5k samples to create the *Finance* dataset, designed to evaluate the model's specialized performance in financial QA tasks.

### B.2. InstructionBench

InstructionBench aims to assess the adaptability and performance of models across a diverse range of general instruction tasks, including, but not limited to, question answering (QA), text summarization, and classification. This benchmark integrates three carefully curated high-quality datasets: Alpaca-GPT4, Dolly, and InstructionWild, encompassing a variety of instruction tasks generated through both human and model-driven approaches. The evaluation is designed to be both comprehensive and rigorous. The distribution of dataset samples and an example table of dataset entries are provided in Figure 6 and Table 8.

**Dolly**: The Dolly-15k[5] dataset, created by Databricks, consists of 15k high-quality, human-generated prompt-response pairs. It is specifically designed for the instruction fine-tuning of large language models. Unlike datasets generated through model outputs or copy-pasting, Dolly-15k maintains authenticity and high quality by relying solely on human input. The dataset encompasses common instruction fine-tuning tasks, including QA, summarization, and classification. We follow the official guide from Databricks[6] and concatenate the sample fields into complete training samples. A subset of 5k samples is randomly selected to evaluate model performance.

**Alpaca-GPT4**: The Alpaca-GPT4[7] dataset comprises 52k instruction-following samples generated using GPT-4. The

---

[3]https://huggingface.co/datasets/wangrongsheng/GenMedGPT-5k-en
[4]https://huggingface.co/datasets/gbharti/wealth-alpaca_lora
[5]https://huggingface.co/datasets/databricks/databricks-dolly-15k
[6]https://github.com/databrickslabs/dolly/blob/master/training/consts.py
[7]https://huggingface.co/datasets/llamafactory/alpaca_gpt4_en

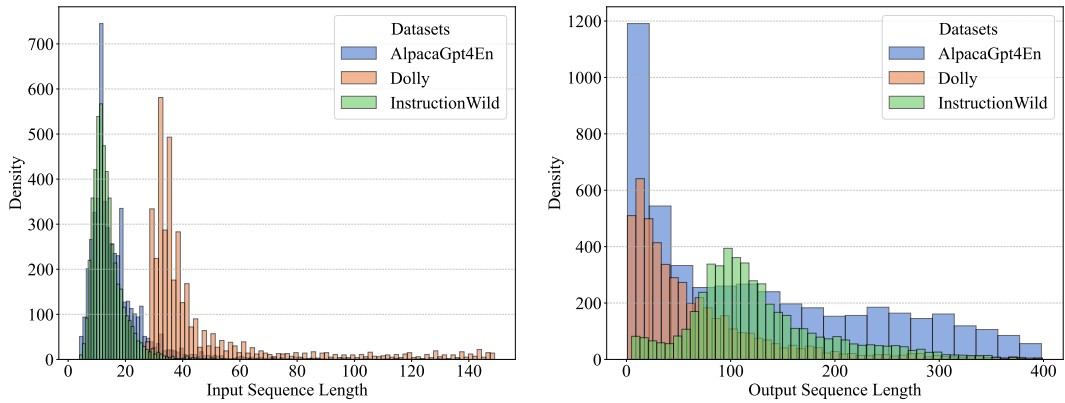

*Figure 6.* Distribution of Sequence Lengths for Samples in InstructionBench.

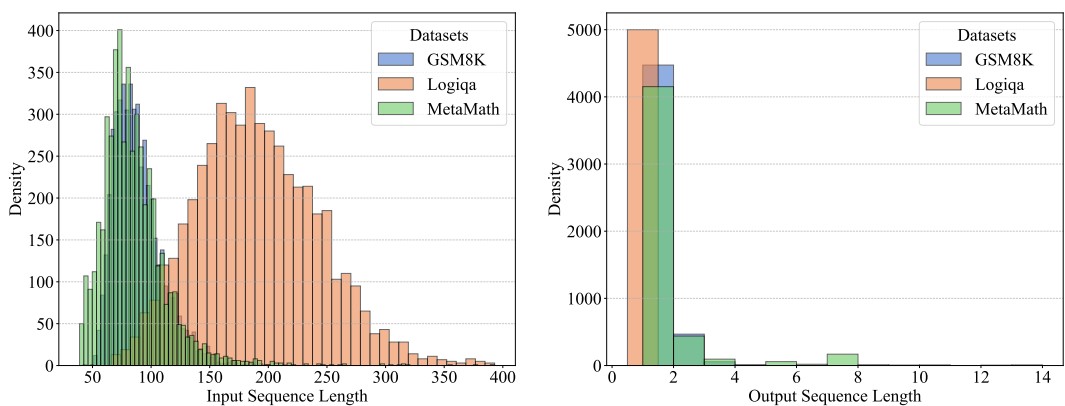

*Figure 7.* Distribution of Sequence Lengths for Samples in ReasoningBench.

dataset is constructed by first manually creating a comprehensive set of instructions across a wide range of tasks, followed by data generation and quality assurance using GPT-4. It includes diverse task types, such as various QA and summarization tasks. From this dataset, we randomly select 5k samples to test the model's generalization capabilities and performance on instruction tasks.

**InstructionWild**: InstructionWild[8] is a large dataset focused on real-world user instructions, consisting of over 50k high-quality bilingual (Zh-En) instructions. The data is derived from real user-shared scenarios, offering diversity and broad applicability. Its format aligns with that of the Alpaca dataset, enabling seamless integration and usage. The dataset covers common instruction fine-tuning tasks such as QA and summarization. We randomly extract 5k samples to evaluate the model's ability to understand and execute instructions effectively.

### B.3. ReasoningBench

ReasoningBench is designed to evaluate models' logical reasoning and problem-solving abilities through tasks such as mathematical problem solving, multi-step reasoning, and logical reading comprehension. This benchmark integrates three high-quality reasoning datasets, GSM8K, MetaMath, and LogiQA, to comprehensively assess models' reasoning performance across diverse dimensions and task types. The distribution of dataset samples and an example table of dataset entries are provided in Figure 7 and Table 9.

---

[8]https://huggingface.co/datasets/fuliucansheng/InstructionWild

*Table 8.* Examples of InstructionBench data.

| | |
|---|---|
| Dolly | **Instruction:** Below is an instruction that describes a task. Write a response that appropriately completes the request.
**Input:** What athlete created the 'beast quake' for the Seattle Seahawks?
**Response:**
**Output:** Marshan Lynch |
| Alpaca-GPT4 | **Input:** Imagine you are talking to a child, explain the concept of DNA.
**Output:** Okay kiddo, so you know how we all have things that make us special and unique? Like how you have your mom's eyes and your dad's hair? Well, that's because of something called DNA. DNA is like the directions or the recipe that our bodies follow to make us who we are. It decides things like the color of our eyes, how tall we'll grow, and even how our personality is.\n\n Just like you follow instructions to build a Lego castle, the cells in our body follow the DNA's instructions to make us who we are. You might also hear DNA being called our "genetic code," because it's kind of like the secret language that our bodies use to build us.\n\n And the cool thing about DNA is that it's passed down from our parents. So, we get half of our DNA from our mom and half from our dad. That's why we can have traits from both of them. But even though we share DNA with our family, while there is nobody else in the world with the exact same DNA as us. We are all unique and special in our own way!\n\n Isn't that cool? Do you have any more question about it? |
| InstructionWild | **Input:** Create a list of five unique travel destinations that are not popular tourist spots. Provide a brief description for each destination, highlighting what sets it apart from other popular travel locations.
**Output:** Meghalaya, India: Located in the northeast part of India, Meghalaya is known for its beautiful rivers, ancient caves, and rich cultural heritage. It\u2019s one of the least explored regions of the country, making it an ideal destination for those looking for an escape from the crowds of popular tourist spots.\n Atacama Desert, Chile: The driest desert in the world, the Atacama Desert is an other-worldly landscape of stunning terrain and amazing geological formations. It\u2019s home to some of the clearest skies on the planet and offers incredible stargazing opportunities.\n Antarctica: For the adventurous traveler, Antarctica is a wild, unforgiving landscape like no other. It\u2019s home to breathtaking glaciers, unspoiled coastlines, and an abundance of wildlife. With few visitors, it\u2019s a paradise for those who want to explore the continent\u2019s remote beauty.\n Georgia: Located between Europe and Asia, Georgia is a little-known country full of culture, history, and stunning landscapes. From the city of Tbilisi to the mountains of Svaneti, there are endless opportunities for exploration and discovery.\n Australian Outback: Few tourists venture to the remote stretches of the Australian Outback, but it\u2019s well worth the effort. Here, you\u2019ll find miles of unspoiled nature, rooted in Indigenous history and culture, and the chance to experience a whole different side of the continent. |

**GSM8K**: GSM8K[9] is a high-quality dataset of linguistically diverse elementary school math word problems, constructed by OpenAI. It contains a total of 8.5k samples. The dataset's unique feature lies in its human-crafted problems, avoiding templated language, and providing natural language solutions instead of purely mathematical expressions. This design ensures a closer alignment with real-world problem-solving scenarios. The primary task type is multi-step mathematical reasoning, which effectively diagnoses deficiencies in a model's reasoning capabilities. Consistent with common pratice, we apply a zero-shot chain-of-thought (CoT) prompt to each sample, guiding the model to think step by step. For evaluation, we combine the training and test sets and randomly select 5k samples.

**MetaMath**: MetaMath[10] is a large-scale dataset comprising approximately 395k samples, designed to enhance mathematical reasoning through a question-guided approach. It diversifies mathematical problems by rephrasing and restructuring them from multiple perspectives, offering a robust and challenging benchmark for evaluating mathematical reasoning abilities. The dataset focuses on QA tasks and spans a broad spectrum of mathematical problem complexities. Similar to GSM8k, we apply a zero-shot CoT prompt to each sample to guide the model in logical reasoning. For evaluation purposes, we randomly select 5k samples from the training set.

---

[9]https://huggingface.co/datasets/openai/gsm8k
[10]https://huggingface.co/datasets/meta-math/MetaMathQA

*Table 9.* Examples of ReasoningBench data.

| | |
|---|---|
| GSM8K | **Input:** Below is an instruction that describes a task. Write a response that appropriately completes the request.\n\n### Instruction:\nFive food companies sponsored a local food bank. Foster Farms donated 45 dressed chickens; American Summits donated twice the number of bottled water than the number of dressed chicken donated by Foster Farms; Hormel donated three times the number of dressed chickens that Foster Farms donated; Boudin Butchers donated one-third of the number of dressed chickens that Hormel donated; Del Monte Foods donated 30 fewer bottles of water than American Summits. How many food items did the companies donate in total? Response: Let's think step by step.
**Output:** 375 |
| MetaMath | **Input:** Below is an instruction that describes a task. Write a response that appropriately completes the request. Instruction:\n What is the value of the ceiling function applied to $\sqrt{\frac{49}{4}}$ ? \n Response: Let's think step by step.
**Output:** -3 |
| LogiQA | **Input:** Write a multi-choice question for the following article:\n Article: Researchers believe that if mothers are exposed to more pesticides in the first few months of pregnancy, the babies born may be less intelligent. They believe that the embryonic brain begins to develop shortly after pregnancy, so the pre-pregnancy is the baby's brain In the critical period of development, exposure to more pesticides may change the environment around the developing embryo's brain in pregnant women.\n Question: Which of the following, if true, would best support a researcher's point of view?\n Options: A. Many babies are born early due to their mothers' exposure to pesticides.\n B. Insecticides are a potential threat to people's health, and it can also cause many diseases such as Parkinson's disease, cancer and mental illness.\n C. Previous research has found that increased exposure to pesticides can cause thyroid problems in pregnant women, and the thyroid status of pregnant women can affect the intellectual development of the fetus.\n D. Researchers conducted a follow-up survey of 1,500 pregnant women and found that children born to pregnant women who were more exposed to pesticides performed significantly worse in mathematics and language.\n\n Answer:
**Output:** C |

**LogiQA**: LogiQA[11] is a high-quality, comprehensive dataset focused on logical reasoning, derived from logical reasoning questions used in the Chinese National Civil Service Examination. It consists of 8k QA samples, covering a variety of deductive reasoning tasks designed to test a model's adaptability to logical reasoning and problem-solving. During dataset construction, strict filtering was applied to exclude samples with inappropriate formats or those involving charts and mathematical calculations, ensuring the dataset's purity and quality. Following the official dataset guidelines[12], we create multiple-choice prompts and randomly select 5k samples for evaluation.

## C. More Details for Experiment Settings

### C.1. More Metrics

To provide a more comprehensive evaluation of the proposed method, we employ additional evaluation metrics, including BERTScore (BS F1) (Zhang et al., 2019), BLEU (Papineni et al., 2002), Rouge-1 (R-1), Rouge-2 (R-2), and Rouge-L (R-L). These metrics help assess various aspects of model performance, such as the quality of generated text, its similarity to reference outputs, and the model's ability to capture key information at different levels of granularity.

### C.2. Implementation Details

The training and evaluation are conducted on the 80G memory-sized NVIDIA A800 GPUs with CUDA version 12.1. Our method is implemented using the PyTorch framework with Pytorch, version 2.5.1. The training framework used is LLaMA-Factory[13]. For the LoRA setup, we use random Gaussian initialization for matrix $\mathcal{A}$ and set matrix $\mathcal{B}$ to zero, with

---

[11]https://huggingface.co/datasets/lucasmccabe/logiqa

[12]https://github.com/csitfun/LogiQA2.0/blob/main/logiqa/multi-choice-prompt.py

[13]https://github.com/hiyouga/LLaMA-Factory

*Table 10.* Comparison of experimental results on the **Geography** dataset of DomainBench.

| Method | BERTScore ↑ | BLEURT ↑ | BLEU ↑ | Rouge-1 ↑ | Rouge-2 ↑ | Rouge-L ↑ |
|---|---|---|---|---|---|---|
| Llama3.2-3B-Instruct | 0.6905 | **-0.6794** | 0.0638 | 0.2661 | 0.1020 | 0.1917 |
| • Tent | 0.6661 | -1.1320 | 0.0444 | 0.2149 | 0.1149 | 0.1780 |
| • EATA | 0.5274 | -1.4276 | 0.0032 | 0.0065 | 0.0007 | 0.0064 |
| • TLM (Ours) | **0.7160** | -0.7278 | **0.0952** | **0.3203** | **0.1526** | **0.2534** |
| Llama3-8B-Instruct | 0.6953 | -0.6161 | 0.0709 | 0.2711 | 0.1040 | 0.1977 |
| • Tent | 0.6007 | -1.5575 | 0.0087 | 0.0932 | 0.0100 | 0.0756 |
| • EATA | 0.6751 | -1.0290 | 0.0575 | 0.2432 | 0.1290 | 0.1949 |
| • TLM (Ours) | **0.7284** | **-0.5860** | **0.1064** | **0.3540** | **0.1650** | **0.2835** |
| Llama2-13B-chat | 0.6707 | -0.7337 | 0.0495 | 0.2430 | 0.0975 | 0.1706 |
| • Tent | 0.4541 | -1.2410 | 0.0011 | 0.0321 | 0.0000 | 0.0320 |
| • EATA | **0.6999** | -0.7661 | 0.0703 | **0.3132** | **0.1441** | **0.2421** |
| • TLM (Ours) | 0.6902 | **-0.6738** | **0.0722** | 0.2996 | 0.1319 | 0.2225 |
| Qwen2.5-7B-Instruct | 0.7003 | -0.5802 | 0.0823 | 0.2911 | 0.1125 | 0.2128 |
| • Tent | 0.6925 | -0.9422 | 0.0810 | 0.2703 | 0.1293 | 0.2242 |
| • EATA | 0.6856 | -0.9806 | 0.0723 | 0.2444 | 0.1182 | 0.2063 |
| • TLM (Ours) | **0.7214** | **-0.5093** | **0.1039** | **0.3412** | **0.1481** | **0.2632** |

a rank of $r = 8$. The LoRA is applied only to the $W_q$ and $W_v$.

Tent (Wang et al., 2021) is a Test-Time Adaptation (TTA) method originally designed for image classification tasks, which adapts a model's parameters during inference based on the entropy of predictions. In the context of dynamic parameter updates for Large Language Models (LLMs), we adapt the Tent method to work with LLMs by leveraging the prediction entropy of the 80 tokens generated by the model. Specifically, we calculate the entropy of the model's predictions for these 80 tokens during each test-time update and use this information to adjust the LLM's parameters.

EATA (Niu et al., 2022a) is a state-of-the-art TTA method for image classification models, which adjusts model parameters based on low-entropy samples during inference. In the context of dynamic parameter updates for LLMs, we adapt the EATA method by leveraging the prediction entropy of 80 tokens generated by the model to select samples. Specifically, during each test-time update, we compute the entropy of the model's predictions for these 80 tokens and, following the setup of EATA, adjust the parameters of the LLM (with the hyperparameter $E_0$ of EATA set to 0.4). However, performing EATA updates on Llama2-13B-chat may result in out-of-memory errors. To address this issue, we reduce the number of tokens from 80 to 30 when applying EATA updates on Llama2-13B-chat.

**Offline and Online Settings.** In our experiments, we consider two distinct settings: Offline and Online. In the Offline setting, all test data is processed at once, and the model's parameters are updated using all available test samples before any testing is performed. In the Online setting, test data sequentially, where the model is updated after each individual test sample or batch. This setting better reflects real-world scenarios where data arrives in a continuous stream, requiring real-time updates to the model.

## D. More Results of Experiment

To comprehensively evaluate the effectiveness of the proposed TLM, we report additional evaluation metrics and results. As shown in Table 10, the proposed method outperforms both the entropy-minimization-based method (Tent) and the original LLMs on the Geography dataset. Specifically, the proposed method achieves a 4.76% improvement in BERTScore compared to Llama3-8B-Instruct. As shown in Table 11, our proposed TLM outperforms Tent on the Agriculture dataset. For instance, our proposed TLM achieves a 146.61% improvement in the BLEU metric compared to Qwen2.5-7B-Instruct. From Table 12, the proposed method outperforms the original LLM on the Medicine dataset. Specifically, compared to Llama3-8B-Instruct, the proposed method achieves a relative improvement of 7.05% in BERTScore. As shown in Table 13, our proposed TLM outperforms Tent on the Finance dataset. For instance, our proposed TLM achieves a relative improvement 122.50% improvement in the BLEU metric compared to Llama3.2-3B-Instruct. From Table 14, the proposed method outperforms the original LLM on the Alpaca-GPT4 dataset. Specifically, compared to Llama3-8B-Instruct, the proposed method achieves a relative improvement of 4.30% in BERTScore. As shown in Table 15, the proposed method

*Table 11.* Comparison of experimental results on the **Agriculture** dataset of DomainBench.

| Method | BERTScore ↑ | BLEURT ↑ | BLEU ↑ | Rouge-1 ↑ | Rouge-2 ↑ | Rouge-L ↑ |
|---|---|---|---|---|---|---|
| Llama3.2-3B-Instruct | 0.6672 | -0.7595 | 0.0104 | 0.0927 | 0.0316 | 0.0684 |
| • Tent | 0.6350 | -0.9975 | 0.0015 | 0.0159 | 0.0034 | 0.0148 |
| • EATA | 0.6659 | -0.8355 | 0.0022 | 0.0263 | 0.0046 | 0.0247 |
| • TLM (Ours) | **0.6787** | **-0.6668** | **0.0288** | **0.1935** | **0.0655** | **0.1518** |
| Llama3-8B-Instruct | 0.6666 | -0.7288 | 0.0106 | 0.0903 | 0.0328 | 0.0665 |
| • Tent | 0.5746 | -1.2577 | 0.0008 | 0.0075 | 0.0011 | 0.0066 |
| • EATA | 0.5870 | -1.3074 | 0.0003 | 0.0019 | 0.0000 | 0.0016 |
| • TLM (Ours) | **0.6526** | **-0.6703** | **0.0189** | **0.1477** | **0.0496** | **0.1110** |
| Llama2-13B-chat | 0.6235 | -0.6134 | **0.0116** | 0.0910 | 0.0319 | 0.0668 |
| • Tent | 0.5901 | -0.9157 | 0.0015 | 0.0200 | 0.0019 | 0.0193 |
| • EATA | 0.6065 | -0.6084 | 0.0101 | 0.0843 | 0.0257 | 0.0675 |
| • TLM (Ours) | **0.6354** | **-0.5929** | 0.0144 | **0.1108** | **0.0379** | **0.0826** |
| Qwen2.5-7B-Instruct | 0.6562 | -0.7226 | 0.0118 | 0.1068 | 0.0377 | 0.0768 |
| • Tent | 0.6667 | -1.1641 | 0.0118 | 0.1399 | 0.0346 | 0.1145 |
| • EATA | 0.6562 | -1.1517 | 0.0113 | 0.1425 | 0.0357 | 0.1166 |
| • TLM (Ours) | **0.6755** | **-0.6219** | **0.0291** | **0.1841** | **0.0663** | **0.1394** |

outperforms both Tent and the original LLMs across all metrics on the Dolly dataset. Specifically, the proposed method achieves a relative improvement of 1.11% in BERTScore compared to Llama3-8B-Instruct. As shown in Table 16, our proposed TLM outperforms Tent on the InstructionWild dataset. For instance, our proposed TLM achieves a relative improvement 50.15% improvement in the BLEU metric compared to Llama3.2-3B-Instruct.

# E. Discussions and Future Works

To the best of our knowledge, addressing the challenges faced by LLMs in real-world deployments, such as distributional shifts in test data, has not been thoroughly explored. In this work, we introduce the Test-Time Learning (TTL) task, aiming to dynamically adapt LLMs using only unlabeled test data during testing. Additionally, we propose the AdaptEval benchmark, designed to evaluate the effectiveness of TTL in enhancing model performance across a variety of domains. Experimental results demonstrate that the proposed Test-Time Learning approach effectively improves LLM performance on target domains. However, we believe there are several potential research directions worth exploring in the future:

**Cross-Domain Continuous Adaptation**: When deploying LLMs across multiple domains, it is essential to achieve continuous adaptation without overfitting to a specific domain. This requires balancing the transfer of knowledge across domains while mitigating catastrophic forgetting. Future work could explore methods for seamless domain adaptation that improves model generalization across dynamic and diverse tasks.

**Only Forward Passes**: LLMs have large parameter sizes, and deployed models in practical settings may not support backpropagation due to memory or computational limitations. This restricts the ability to perform Test-Time Learning during inference, highlighting the need for more efficient methods that enable model adaptation without requiring backpropagation.

*Table 12.* Comparison of experimental results on the **Medicine** dataset of DomainBench.

| Method | BERTScore ↑ | BLEURT ↑ | BLEU ↑ | Rouge-1 ↑ | Rouge-2 ↑ | Rouge-L ↑ |
|---|---|---|---|---|---|---|
| Llama3.2-3B-Instruct | 0.6677 | -0.7668 | 0.0165 | 0.1588 | 0.0269 | 0.1037 |
| • Tent | 0.6322 | -1.2819 | 0.0191 | 0.1897 | 0.0276 | 0.1204 |
| • EATA | 0.6554 | -0.9669 | 0.0018 | 0.0312 | 0.0017 | 0.0178 |
| • TLM (Ours) | **0.7005** | **-0.5603** | **0.0463** | **0.2559** | **0.0728** | **0.1844** |
| Llama3-8B-Instruct | 0.6628 | -0.7486 | 0.0136 | 0.1398 | 0.0343 | 0.0911 |
| • Tent | 0.5525 | -1.3487 | 0.0014 | 0.0113 | 0.0006 | 0.0089 |
| • EATA | 0.6072 | -1.1079 | 0.0011 | 0.0139 | 0.0011 | 0.0100 |
| • TLM (Ours) | **0.7095** | **-0.4781** | **0.0486** | **0.2646** | **0.0836** | **0.1889** |
| Llama2-13B-chat | 0.6559 | -0.5971 | 0.0158 | 0.1439 | 0.0397 | 0.0956 |
| • Tent | 0.6235 | -0.5867 | 0.0100 | 0.1250 | 0.0261 | 0.0874 |
| • EATA | 0.6543 | -0.4349 | 0.0145 | 0.1465 | 0.0410 | 0.1007 |
| • TLM (Ours) | **0.6988** | **-0.4615** | **0.0512** | **0.2370** | **0.0890** | **0.1760** |
| Qwen2.5-7B-Instruct | 0.6561 | -0.6598 | 0.0136 | 0.1441 | 0.0369 | 0.0909 |
| • Tent | 0.5884 | -1.1911 | 0.0097 | 0.0548 | 0.0214 | 0.0454 |
| • EATA | 0.6408 | -0.7171 | 0.0163 | 0.1672 | 0.0304 | 0.1202 |
| • TLM (Ours) | **0.7082** | **-0.5964** | **0.0623** | **0.2623** | **0.1003** | **0.1967** |

*Table 13.* Comparison of experimental results on the **Finance** dataset of DomainBench.

| Method | BERTScore ↑ | BLEURT ↑ | BLEU ↑ | Rouge-1 ↑ | Rouge-2 ↑ | Rouge-L ↑ |
|---|---|---|---|---|---|---|
| Llama3.2-3B-Instruct | 0.6809 | -0.6532 | 0.0360 | 0.2425 | 0.0791 | 0.1602 |
| • Tent | 0.6322 | -1.2819 | 0.0191 | 0.1149 | 0.0549 | 0.1084 |
| • EATA | 0.5028 | -1.2700 | 0.0010 | 0.0150 | 0.0001 | 0.0149 |
| • TLM (Ours) | **0.7060** | **-0.5113** | **0.0801** | **0.3206** | **0.1255** | **0.2367** |
| Llama3-8B-Instruct | 0.6862 | -0.6230 | 0.0407 | 0.2530 | 0.0873 | 0.1682 |
| • Tent | 0.5919 | -0.9704 | 0.0032 | 0.0390 | 0.0072 | 0.0351 |
| • EATA | 0.6431 | -1.2160 | 0.0149 | 0.1398 | 0.0567 | 0.1183 |
| • TLM (Ours) | **0.7153** | **-0.4411** | **0.0952** | **0.3514** | **0.1448** | **0.2576** |
| Llama2-13B-chat | 0.6814 | -0.6217 | 0.0427 | 0.2591 | 0.0892 | 0.1722 |
| • Tent | 0.4878 | -1.1973 | 0.0006 | 0.0049 | 0.0000 | 0.0049 |
| • EATA | 0.6203 | -1.0560 | 0.0250 | 0.1241 | 0.0574 | 0.1113 |
| • TLM (Ours) | **0.6979** | **-0.5110** | **0.0628** | **0.3009** | **0.1104** | **0.2064** |
| Qwen2.5-7B-Instruct | 0.6978 | -0.5370 | 0.0639 | 0.2969 | 0.1124 | 0.2025 |
| • Tent | 0.6623 | -1.1316 | 0.0313 | 0.1835 | 0.0845 | 0.1533 |
| • EATA | 0.7103 | -0.5746 | 0.0810 | 0.3127 | 0.1358 | 0.2409 |
| • TLM (Ours) | **0.7220** | **-0.3890** | **0.0968** | **0.3607** | **0.1487** | **0.2616** |

*Table 14.* Comparison of experimental results on the **Alpaca-GPT4** dataset of InstructionBench.

| Method | BERTScore ↑ | BLEURT ↑ | BLEU ↑ | Rouge-1 ↑ | Rouge-2 ↑ | Rouge-L ↑ |
|---|---|---|---|---|---|---|
| Llama3.2-3B-Instruct | 0.7260 | -0.5164 | 0.0953 | 0.3885 | 0.1712 | 0.2619 |
| • Tent | 0.5656 | -1.5454 | 0.0054 | 0.0351 | 0.0032 | 0.0305 |
| • EATA | 0.6553 | -1.2411 | 0.0220 | 0.1513 | 0.0753 | 0.1359 |
| • TLM (Ours) | **0.7523** | **-0.4148** | **0.1239** | **0.4170** | **0.2062** | **0.3184** |
| Llama3-8B-Instruct | 0.7353 | -0.4655 | 0.1082 | 0.4076 | 0.1863 | 0.2796 |
| • Tent | 0.6712 | -1.0652 | 0.0421 | 0.2193 | 0.1193 | 0.1881 |
| • EATA | 0.6398 | -0.8220 | 0.0271 | 0.1499 | 0.0725 | 0.1332 |
| • TLM (Ours) | **0.7669** | **-0.3090** | **0.1479** | **0.4582** | **0.2349** | **0.3528** |
| Llama2-13B-chat | 0.7326 | -0.5290 | 0.1091 | 0.4068 | 0.1830 | 0.2777 |
| • Tent | 0.6249 | -1.4903 | 0.0108 | 0.0998 | 0.0444 | 0.0937 |
| • EATA | 0.5073 | -0.7398 | 0.0094 | 0.0813 | 0.0191 | 0.0799 |
| • TLM (Ours) | **0.7386** | **-0.5150** | **0.1229** | **0.4311** | **0.1987** | **0.3040** |
| Qwen2.5-7B-Instruct | 0.7624 | -0.2949 | 0.1556 | 0.4803 | 0.2353 | 0.3406 |
| • Tent | 0.6787 | -0.9809 | 0.0501 | 0.2323 | 0.1315 | 0.2031 |
| • EATA | 0.0000 | -1.9681 | 0.0000 | 0.0000 | 0.0000 | 0.0000 |
| • TLM (Ours) | **0.7829** | **-0.2137** | **0.1749** | **0.4919** | **0.2648** | **0.3819** |

*Table 15.* Comparison of experimental results on the **Dolly** dataset of InstructionBench.

| Method | BERTScore ↑ | BLEURT ↑ | BLEU ↑ | Rouge-1 ↑ | Rouge-2 ↑ | Rouge-L ↑ |
|---|---|---|---|---|---|---|
| Llama3.2-3B-Instruct | 0.7289 | -0.4824 | 0.0946 | 0.3778 | 0.1903 | 0.2935 |
| • Tent | 0.6866 | -0.5922 | 0.0797 | 0.2374 | 0.1398 | 0.2039 |
| • EATA | 0.5767 | -1.5690 | 0.0027 | 0.0092 | 0.0002 | 0.0086 |
| • TLM (Ours) | **0.7334** | **-0.4783** | **0.1030** | **0.3878** | **0.1997** | **0.3048** |
| Llama3-8B-Instruct | 0.7415 | -0.4088 | 0.1114 | 0.4126 | 0.2137 | 0.3222 |
| • Tent | 0.5905 | -1.4648 | 0.0000 | 0.0036 | 0.0000 | 0.0036 |
| • EATA | 0.6559 | -1.2475 | 0.0275 | 0.1991 | 0.0868 | 0.1661 |
| • TLM (Ours) | **0.7497** | **-0.3972** | **0.1194** | **0.4239** | **0.2226** | **0.3388** |
| Llama2-13B-chat | 0.7074 | -0.8015 | 0.0743 | 0.3274 | 0.1550 | 0.2407 |
| • Tent | 0.4111 | -0.7963 | 0.0008 | 0.0076 | 0.0000 | 0.0076 |
| • EATA | 0.5355 | -0.9809 | 0.0076 | 0.0632 | 0.0175 | 0.0577 |
| • TLM (Ours) | **0.7108** | **-0.8169** | **0.0789** | **0.3411** | **0.1625** | **0.2516** |
| Qwen2.5-7B-Instruct | 0.7212 | -0.4784 | 0.0911 | 0.3501 | 0.1790 | 0.2634 |
| • Tent | 0.6674 | -1.1697 | 0.0420 | 0.2373 | 0.1036 | 0.1051 |
| • EATA | 0.6737 | -1.1244 | 0.0491 | 0.2482 | 0.1132 | 0.1963 |
| • TLM (Ours) | **0.7216** | **-0.5209** | **0.0931** | **0.3567** | **0.1832** | **0.2685** |

*Table 16.* Comparison of experimental results on the **InstructionWild** dataset of InstructionBench.

| Method | BERTScore ↑ | BLEURT ↑ | BLEU ↑ | Rouge-1 ↑ | Rouge-2 ↑ | Rouge-L ↑ |
|---|---|---|---|---|---|---|
| Llama3.2-3B-Instruct | 0.7019 | -0.4837 | 0.0337 | 0.2796 | 0.0942 | 0.1699 |
| • Tent | 0.5409 | -1.2307 | 0.0018 | 0.0341 | 0.0008 | 0.0341 |
| • EATA | 0.4537 | -1.3202 | 0.0011 | 0.0123 | 0.0006 | 0.0122 |
| •TLM (Ours) | **0.7044** | **-0.4503** | **0.0506** | **0.3104** | **0.1136** | **0.2004** |
| Llama3-8B-Instruct | 0.7071 | -0.4469 | 0.0363 | 0.2846 | 0.0976 | 0.1735 |
| • Tent | 0.6092 | -0.5438 | 0.0148 | 0.0933 | 0.0317 | 0.0910 |
| • EATA | 0.6455 | -0.5337 | 0.0133 | 0.0956 | 0.0452 | 0.1125 |
| • TLM (Ours) | **0.7133** | **-0.3669** | **0.0554** | **0.3206** | **0.1207** | **0.2050** |
| Llama2-13B-chat | 0.7025 | -0.4040 | 0.0464 | 0.3032 | 0.1062 | 0.1860 |
| • Tent | 0.5390 | -0.6001 | 0.0253 | 0.1294 | 0.0464 | 0.1260 |
| • EATA | 0.5869 | -0.7337 | 0.0159 | 0.1129 | 0.0295 | 0.0947 |
| • TLM (Ours) | **0.7056** | **-0.3792** | **0.0499** | **0.3122** | **0.1119** | **0.1926** |
| Qwen2.5-7B-Instruct | 0.7071 | -0.4295 | 0.0436 | 0.3116 | 0.1070 | 0.1875 |
| • Tent | 0.6865 | -1.0060 | 0.0153 | 0.1829 | 0.1232 | 0.1615 |
| • EATA | 0.6911 | -0.9889 | 0.0157 | 0.1830 | 0.1233 | 0.1615 |
| • TLM (Ours) | **0.7261** | **-0.2746** | **0.0813** | **0.3806** | **0.1498** | **0.2472** |

