# OpenReview forum: "Test-Time Learning for Large Language Models"
_ICML.cc/2025/Conference — ICML 2025 poster_

### Official Review · Reviewer_ftxN · 2025-03-05

**Overall Recommendation:** 4

**Summary:**

The paper introduces Test-Time Learning (TTL) paradigm for Large Language Models (LLMs), termed TLM, designed to dynamically adapt LLMs to target domains using only unlabeled test data during inference. The authors propose three key components: (1) an input perplexity minimization objective, based on empirical and theoretical evidence that reducing input perplexity enhances autoregressive predictions; (2) a Sample Efficient Learning Strategy, which prioritizes high-perplexity samples for efficient model updates; and (3) the use of Low-Rank Adaptation (LoRA) to mitigate catastrophic forgetting and enable lightweight updates. The paper also introduces the AdaptEval benchmark, comprising DomainBench, InstructionBench, and ReasoningBench, to evaluate TTL across diverse tasks. Experimental results claim at least a 20% performance improvement over baseline LLMs on domain knowledge adaptation, with further gains in instruction-following and reasoning tasks, validated on models like Llama3.1-8B-Instruct and Qwen2.5-7B-Instruct.

**Claims And Evidence:**

The claims are generally supported by clear and convincing evidence, including empirical observations, theoretical justifications, and experimental results.

The core claim—that minimizing input perplexity improves LLM performance—is backed by Observation 1 (Fig. 1b)

The claim that high-perplexity samples are more informative is supported by Observation 2 (Fig. 1c)

Experimental results (Tables 2, 3, 4) consistently show TLM outperforming baselines (e.g., Tent, EATA) and original LLMs, with a reported 20%+ improvement on DomainBench

**Essential References Not Discussed:**

N/A

**Experimental Designs Or Analyses:**

I reviewed the experimental designs in Sec. 5 and Supp. D. The comparison experiments (Tables 2, 3) against baselines (Tent, EATA) and original LLMs are well-designed, using diverse models and datasets to test robustness.

One experiment question is that how does your methods compared with other adapting methods such as fine tuning and rag.

**Methods And Evaluation Criteria:**

The proposed TLM method—combining perplexity minimization, sample-efficient learning, and LoRA—makes sense for adapting LLMs to distribution shifts without labeled data, aligning with the problem of real-world deployment challenges.

The use of perplexity as an unsupervised objective leverages LLMs’ autoregressive nature, which is a conceptually sound departure from entropy-based TTA methods ill-suited for LLMs.

**Other Comments Or Suggestions:**

1. Consider moving supplementary metrics (e.g., BERTScore) to the main paper for a richer evaluation.

2. Expand the discussion of trade-offs in the online setting (e.g., adaptation quality vs. efficiency).

**Other Strengths And Weaknesses:**

strength:
1. The paper is well-structured, with clear figures (e.g., Fig. 1) and a logical flow from theory to experiments.
2. The focus on unlabeled test-time adaptation and the AdaptEval benchmark tackle real-world LLM limitations.

weakness:
1. Reproducibility: Missing details (e.g., random seeds, dataset splits) hinder replication.

**Questions For Authors:**

I did not quite understand the Eq 4, why Minimizing the perplexity to the input P(x; Θ) is equivalent to maximizing the input generation probability P(x; Θ).

**Relation To Broader Scientific Literature:**

The paper builds on prior work in test-time methods for classification. The perplexity minimization objective extends autoregressive modeling principles, shifting them to test-time adaptation.

**Theoretical Claims:**

I checked the correctness of the theoretical justification for input perplexity minimization (Sec. 4.1, Eqns. 4-7).

The argument has on two assumptions: (1) the autoregressive property of LLMs and (2) shared parameter influence on input and output perplexity.

---

> ### Author Rebuttal · Authors · 2025-04-01
>
> We are deeply grateful for your thoughtful and encouraging feedback. Your recognition of our motivation and the thoroughness of our experiments is truly inspiring. Our detailed responses are as follows:
>
> >Q1. One experiment question is that how does your methods compared with other adapting methods such as fine tuning and rag.
>
> **A1.** Thank you for the insightful question. Our work focuses on TTL, which adapts LLMs using only *unlabeled test data* during inference. In our experiments, we thus chose to compare our method with other TTA methods, which share a similar problem setting. We would like to **clarify further the rationale for not including fine-tuning and RAG-based methods in our experimental comparison, along with the potential advantages of our method**:
> * **Supervision and Data Dependency.** Fine-tuning requires labeled data, which is costly and often impractical in deployment. RAG depends on a well-maintained retrieval corpus, which may not generalize well across domains. In contrast, **our method uses only test data and minimizes perplexity to adapt efficiently without labels or external knowledge**.
> * **Experimental Design Rationale.** Our experimental design is deliberately aligned with the TTL setting, where no labels or external knowledge sources are available at deployment. Therefore, we select baselines (e.g., Tent, EATA) that also operate in this setting. **Although fine-tuning and RAG are powerful in their respective contexts, they fall outside the scope of our setting due to their dependency on supervised data or retrieval infrastructure.**
>
> >Q2. Reproducibility: Missing details (e.g., random seeds, dataset splits) hinder replication.
>
> **A2.** We appreciate your feedback on ensuring reproducibility. Below we clarify the missing details and emphasize the existing descriptions in our paper:
> * **Random Seeds.** Ours experiments use a fixed random seed of **42** for PyTorch/Numpy initialization.
> * **Dataset Splits for AdaptEval Benchmark.** The proposed AdaptEval benchmark is specifically designed for TTL. Unlike traditional benchmarks, **AdaptEval contains only test-domain data without predefined train/val/test splits**, as TTL methods adapt solely during inference. Full dataset construction protocols are provided in Supp. B.
> * **Additional Reproducibility Guarantees.** **1)** All hyperparameters (learning rates, batch sizes, adaptation steps) are detailed in Sec. 5.1. **2)** Hardware specifications (e.g., GPU types) and library versions (PyTorch) are detailed in Supp. C.2. **3)** Code, pre-trained models, and AdaptEval data will be publicly released upon acceptance.
>
>
> >Q3. Consider moving supplementary metrics (e.g., BERTScore) to the main paper for a richer evaluation.
>
> **A3.** We appreciate the feedback and agree that incorporating supplementary metrics like BERTScore into the main paper will strengthen the evaluation's comprehensiveness.
>
> >Q4. Expand the discussion of trade-offs in the online setting (e.g., adaptation quality vs. efficiency).
>
> **A4.** Thanks for your valuable comments. We conduct additional experiments in the **online setting** to evaluate the effect of batch size under streaming-like scenarios (see Table C). **With bs=1, the model updates after each test sample, simulating a true streaming setting. While this increases update steps, our method remains effective.** Using bs=100 reduces update frequency while slightly improving performance. This indicates that accumulating more samples before updating provides more reliable learning signals, and thus enhances the adaptation performance. The results suggest that in an online setting, adopting as high a batch size as possible can lead to both effective and efficient performance. The results suggest that in the online scenario, while more frequent updates offer immediate adaptation, adopting a larger batch size can lead to more effective and efficient adaptation.
>
> Table C. Experimental results in the **Online setting** on DomainBench.
> | Methods|Average R-Lsum|#Backwards|
> |-|-|-|
> |Ours (bs=100)|0.2040|1514|
> |Ours (bs=1)|0.1917|2541|
>
> >Q5. I did not quite understand the Eq 4, why Minimizing the perplexity to the input $\mathcal{P}(x;\Theta)$ is equivalent to maximizing the input generation probability $P(x;\Theta)$.
>
> **A5.** Thank you for your insightful suggestion. The equivalence between minimizing input perplexity and maximizing the input generation probability is a fundamental property of perplexity in probabilistic models. We clarify this relationship as follows. Perplexity $\mathcal{P}(x;\Theta)$ for a sequence $x$ is defined as the exponential of the **cross-entropy** loss $P(x;\Theta)$, Formally, Eq. (2) can be written as：$\mathcal{P}(x;\Theta)=e^{-\frac{1}{N}\log P(x;\Theta)}$, where $N$ is the number of tokens in the sequence. Therefore, maximizing $\log P(x;\Theta)$ directly corresponds to minimizing $\mathcal{P}(x;\Theta)$.
> ****
> We sincerely hope our clarifications above have addressed your concerns.

---

### Official Review · Reviewer_LK6b · 2025-03-09

**Overall Recommendation:** 3

**Summary:**

This paper proposes a novel Test-Time Learning (TTL) method that assigns weights to different samples based on input perplexity and employs LoRA for model adaptation. Experimental results demonstrate that, compared with existing TTL methods, the proposed approach achieves superior performance across multiple datasets.

**Claims And Evidence:**

The majority of claims presented in the paper are adequately supported by experimental results. However, regarding the claim of **"Reducing Output Perplexity through Input Perplexity Minimization,"** the theoretical justification provided is somewhat unclear. While the autoregressive property indeed implies predicting subsequent tokens based on previous ones, the input content is inherently given (fixed), whereas the output tokens are generated sequentially. Therefore, it is not theoretically evident that minimizing input perplexity directly implies a corresponding reduction in output perplexity.

**Essential References Not Discussed:**

There are no essential references missing from the paper.

**Experimental Designs Or Analyses:**

The experimental design is reasonably thorough and comprehensive. However, the selection of models is relatively limited. Including models such as LLaMA-2-13B, which might be somewhat outdated, is less convincing. It would strengthen the work to evaluate newer and larger-scale models from multiple model families to demonstrate broader applicability.

**Methods And Evaluation Criteria:**

Overall, the methods and evaluation criteria employed in this work are reliable and sound. Nevertheless, the choice of the threshold $P_0$ for the perplexity-based weighting scheme warrants further investigation. Currently, the authors have empirically tested various values of  $P_0$  only on DomainBench datasets, without assessing its generalizability to reasoning-oriented tasks. Introducing a more generalizable or theoretically justified criterion for selecting $P_0$ would further strengthen the methodology.

**Other Comments Or Suggestions:**

See above.

**Other Strengths And Weaknesses:**

See above.

**Questions For Authors:**

the baselines used for comparison (Tent and EATA) are somewhat outdated, being methods from 2021 and 2022. Considering recent advances summarized in the survey *"A Comprehensive Survey on Test-Time Adaptation under Distribution Shifts"*, are there any newer state-of-the-art test-time learning methods for a more comprehensive evaluation and stronger evidence of the proposed approach’s superiority?

**Relation To Broader Scientific Literature:**

The key contributions of the paper are closely related to the broader scientific literature on large language models and their reasoning capabilities. It addresses the issue of Test-Time Learning.

**Theoretical Claims:**

No theoretical part in this paper.

---

> ### Author Rebuttal · Authors · 2025-04-01
>
> We thank the reviewer for the encouraging comments and detailed suggestions. Responses are below:
>
> >Q1. Regarding the claim of "Reducing Output Perplexity through Input Perplexity Minimization," the theoretical justification provided is somewhat unclear.
>
> **A1.** We appreciate your feedback. We clarify our motivation below:
> * **Theoretical Motivation Clarification.** In our study, we observe the strong positive correlation between input and output PPL (see Fig.1b). Relying on this, we seek to reduce the output PPL by input PPL minimization.
> * **Autoregressive Training Dynamics.** In autoregressive models, each token is generated using the internal representation of the input. Although the model parameters are updated during input PPL minimization, this process refines the representation $h(x;\Theta)$, which in turn informs the generation of $P(y|x;\Theta)$. In other words, since $P(y|x; \Theta)$ is generated directly from $h(x;\Theta)$, an improved representation of $x$ can lead to more accurate and confident next-token predictions, which is expected to reduce output PPL.
> * **Empirical Validation.** We compare the changes in both input and output PPL between our proposed method and the original LLM. In Tab. A, **our method not only reduces input PPL but also leads to a consistent decrease in output PPL**.
>
> Tab. A: Experimental results (metric is PPL) on DomainBench.
> || Methods|Geo.|Agri.|Med.|Fin.|
> |-|-|-|-|-|-|
> |Input PPL|Llama3.1-8B|187.3|933.2|43.6|323.0|
> ||Ours|6.6|3.9|2.0|5.5|
> |Output PPL|Llama3.1-8B|3242.9|205247.1|7.7|1208975.9|
> ||Ours| 2483.9|156232.3|6.6|242617.0|
>
> >Q2. Need to empirically test various values of $P_0$  on reasoning-oriented tasks. Introducing a more generalizable or theoretically justified criterion for selecting $P_0$ would further strengthen the method.
>
> **A2.** We would like to clarify the applicability of the threshold $\mathcal{P}_0$ in reasoning-oriented tasks, and outline potential strategies for its dynamic selection:
> * **Generalization of $\mathcal{P}_0=e^3$ in reasoning-oriented tasks.** We conduct experiments with values of $\mathcal{P}_0=\\{e^2,e^3,e^4\\}$. In Tab. B, when $\mathcal{P}_0=e^3$, our method achieves the best performance on three datasets. These results indicate that $\mathcal{P}_0=e^3$ generalizes well to reasoning-oriented tasks.
> * **Potential method for dynamic threshold selection.** To address the concern regarding the generalizability of the static threshold, one can adopt a dynamic threshold selection scheme. Instead of a fixed $\mathcal{P}_0$, we define $\mathcal{P}_0^{(t)}=\mu\_{\mathcal{P}}^{(t)} + \alpha \cdot \sigma\_{\mathcal{P}}^{(t)}$, where $\mu\_{\mathcal{P}}^{(t)}$ and $\sigma\_{\mathcal{P}}^{(t)}$ are the mean and standard deviation of test samples PPL at step $t$. This method automatically identifies high-surprisal samples critical for adaptation while excluding redundant low-PPL data.
>
> Tab. B: Effects of different $\mathcal{P}_0$ under Llama3.1-8B on ReasoningBench.
> |$\mathcal{P}_0$|$e^2$|$e^3$|$e^4$|
> |-|-|-|-|
> |GSM8K|0.8070|**0.8074**|0.8026|
> |MetaMath|0.7002|**0.7006**|0.7006|
> |Logiqa|0.4834|**0.4868**|0.4538|
>
> >Q3. There is a need to evaluate newer and larger-scale models from multiple model families.
>
> **A3.** We strengthen our empirical validation by expanding experiments across model families and scales. Specifically,
> * **Existing Multi-Family Model Coverage.** As shown in Tables 2-3, our experiments already encompass: **1) Llama Family:** 3B, 8B, and 13B parameter variants. **2) Qwen Family:** Latest Qwen2.5-7B-Instruct model.
> * **New Model Family Added.** To further validate broader applicability, we conduct additional experiments with **Phi-4-14B**. In Tab. C, our method consistently outperforms existing baselines, demonstrating its effectiveness on larger-scale model adaptation tasks.
>
> Tab. C: Experimental results on *DomainBench*.
> | Methods|Geo.|Agri.|Med.|Fin.|
> |-|-|-|-|-|
> |Phi-4-14B|0.2326|0.0997|0.1291|0.2517|
> |Tent|0.0000|0.1178|0.1262|0.2206|
> |EATA|0.0064|0.0143|0.1388|0.2200|
> |Ours|**0.2421**|**0.1315**|**0.1393**|**0.2711**|
>
> >Q4. Need to add SOTA TTA methods for a more comprehensive evaluation.
>
> **A4.** We agree that evaluating against more recent TTA methods can provide a more comprehensive comparison and strengthen the empirical validation of our method. **We include COME [r1], a recently proposed TTA method accepted at ICLR 2025**, in our experimental comparison. In Tab. D, the results demonstrate that **our method consistently and significantly outperforms COME across all domains**.
>
> Tab. D: Comparison between COME and our method on DomainBench.
> | Methods|Geo.|Agri.|Med.|Fin.|
> |-|-|-|-|-|
> |COME [r1]|0.0048|0.0039|0.0301|0.0328|
> |Ours|**0.3212**|**0.1319**|**0.2372**|**0.3242**|
>
> [r1] COME: Test-time Adaption by Conservatively Minimizing Entropy, ICLR 2025.
> ****
> We sincerely hope our clarifications above have addressed your concerns. We would be grateful if you could kindly reconsider the evaluation of our paper.

---

> > ### Comment · Reviewer_LK6b · 2025-04-04
> >
> > Thank you for the rebuttal. I still believe that comparison with more recent TTA methods could improve the work, I appreciate the clarification and will keep my score (Weak Accept).

---

### Official Review · Reviewer_CEYm · 2025-03-10

**Overall Recommendation:** 2

**Summary:**

In this paper, the authors propose a new Test-Time Learning (TTL) approach called Test-Time Learning for LLMs (TLM) that uses unlabeled test data to address distribution shifts arising from specialized domains. They highlight three main contributions:

1. Self-Supervised Perplexity Minimization: The authors set a perplexity-minimization objective to adapt LLMs at test time through a self-supervised approach.
2. Sample-Efficient Learning Strategy: To achieve more efficient and effective parameter updates, they selectively utilize high-perplexity samples for training, thereby focusing on instances that provide the most useful information.
3. LoRA for Stable Adaptation: They integrate LoRA to mitigate catastrophic forgetting and ensure stable model adaptation during test-time learning.

Furthermore, the authors introduce a new benchmark called AdaptEval to comprehensively evaluate TLM. AdaptEval covers vertical domain knowledge, instruction-following tasks, and reasoning datasets, thereby assessing the utility of TLM under various real-world distribution shifts. Experimental results demonstrate that TLM substantially improves performance on data that are highly specialized or deviate from the training distribution compared to existing methods, making it a promising solution in real-world deployment scenarios where labeled data are scarce.

**Claims And Evidence:**

Strengths:
  - The paper convincingly explains why large-scale foundation models struggle to adapt to specialized domains or changing distributions, and provides a solid rationale for the necessity of TLM as a solution.
  - Through both theoretical and empirical evidence on the relationship between perplexity and autoregressive models, the paper strongly justifies the efficacy of perplexity-based optimization. In addition, it clearly shows how LoRA helps mitigate catastrophic forgetting.
  - By utilizing the variety of datasets in the AdaptEval benchmark—including domain knowledge, instruction-following tasks, and reasoning tasks—the paper demonstrates performance improvements of the proposed method in multiple contexts.

Weaknesses:
  - Although the proposed method achieves generally positive experimental results, there are no concrete examples of actual outputs included (only input/output examples of the datasets are mentioned), making it less clear how the model performs in real usage scenarios.
  - The paper compares its method mainly with Tent and EATA, yet there are many other recent TTA methods not discussed. Moreover, while earlier in the paper it references fine-tuning and RAG approaches, a direct comparison to these methods is missing, which is regrettable.

**Essential References Not Discussed:**

- Same as the second weakness in "Claims And Evidence"

**Experimental Designs Or Analyses:**

Strengths:
  - By conducting experiments on the AdaptEval benchmark—which spans a variety of tasks—this design is shown to possess strong robustness.

Weaknesses:
  - Same as the second weakness in "Claims And Evidence"

**Methods And Evaluation Criteria:**

- AdaptEval covers a broad range of distribution shifts—domain knowledge, instruction tasks, and reasoning—so it appears to be well-suited for evaluating both the generality and performance of the proposed method.

**Other Comments Or Suggestions:**

- Overall, this study is well-structured in assuming situations where obtaining high-quality labels is difficult in real operational environments, enabling models to adapt to available test data on the fly.
- It highlights the importance of test-time learning, particularly perplexity-based updates, in large language models, and underscores the practicality of LoRA for mitigating catastrophic forgetting.
- If possible, supplementing the paper with more detailed analyses on extreme cases and computational efficiency, along with comparisons to the latest literature, could further bolster its contributions.

**Other Strengths And Weaknesses:**

Strengths:
  - Combining high-perplexity sample selection with LoRA to balance computational cost and adaptation performance is conceptually neat and practically beneficial.

Weaknesses:
  - More detailed comparisons of time complexity and resource requirements would be welcome when handling different types of domains (e.g., conversational tasks, online streaming data).

**Questions For Authors:**

Same as above.

**Relation To Broader Scientific Literature:**

- Addressing distribution shifts in deployment environments is a crucial issue in the fields of NLP and deep learning, and this work reflects the latest trends, such as perplexity-based updates and the application of LoRA.
- AdaptEval appears likely to serve as a unified standard for subsequent research on TTA, TTT, and similar methods.

**Theoretical Claims:**

- The assumption that reducing the model’s input perplexity also lowers its output perplexity is highly convincing, especially given the autoregressive nature of the model and prior findings in language modeling.

---

> ### Author Rebuttal · Authors · 2025-04-01
>
> We sincerely appreciate your high level of encouragement for our work. Your recognition of "provides **a solid rationale for the necessity of TLM** as a solution", "the paper **strongly justifies the efficacy of perplexity-based optimization**", and "AdaptEval appears likely to **serve as a unified standard for subsequent research**" is deeply appreciated. Our detailed responses are below:
>
> >Q1. Although the proposed method achieves generally positive experimental results, there are no concrete examples of actual outputs included (only input/output examples of the datasets are mentioned), making it less clear how the model performs in real usage scenarios.
>
> **A1.** We sincerely thank you for highlighting the need for concrete examples to demonstrate practical efficacy. As shown in https://anonymous.4open.science/r/ICML1861, our method consistently outperforms SOTA methods across diverse scenarios. Our analysis is detailed below:
> * **More accurate reasoning steps.** In reasoning tasks, our method produces logically sound and well-structured intermediate steps, enabling the model to reach correct conclusions more reliably.
> * **Better coherence and fluency.** Compared to TTA methods, our method maintains output fluency and avoids the repetition issues commonly observed in models like Tent or EATA.
>
> >Q2. Recent TTA methods not discussed. Moreover, a direct comparison to fine-tuning and RAG approaches is missing, which is regrettable.
>
> **A2.** According to your suggestions, we further compare with a SOTA TTA method, COME (ICLR'25,[r1]). From the results in Table A, **our method consistently and significantly outperforms COME across all domains**. This further demonstrate the superior performance of our method in addressing LLMs adaptation to new domains using only test data.
>
> **Experimental Design Rationale with RAG and fine-tuning approaches.** Our experiments focus on TTL, which uses only unlabeled test data without relying on additional labels or retrieval resources. Fine-tuning typically depends on task-specific labeled data, while RAG assumes a well-maintained knowledge base that may not generalize across domains. In contrast, our method updates solely on unlabeled test data via perplexity minimization. Therefore, we compare with TTA methods that share similarly label-free, resource-free settings.
>
> Table A: Comparison between COME and our method on *DomainBench*.
> | Methods|Geo.|Agri.|Med.|Fin.|
> |-|-|-|-|-|
> |COME [r1]|0.0048|0.0039|0.0301|0.0328|
> | **TLM(Ours)** | **0.3212**|**0.1319**| **0.2372** |**0.3242**  |
>
> [r1] COME: Test-time Adaption by Conservatively Minimizing Entropy, ICLR 2025.
>
> >Q3&Q4. More detailed comparisons of time complexity and resource requirements would be welcome when handling different types of domains (e.g., conversational tasks, online streaming data).
>
> **A3&A4.** Thank you for the valuable comments. According to your suggestions, we add experiments under the **online streaming setting (batch size = 1)**, which more accurately reflect real-world inference conditions, such as conversational or sequential inputs. Detailed analyses are as follows:
> *  **Time Complexity.** In the TTL setting, total computation time $T_{\text{total}}$ consists of forward passes plus backward updates, i.e., $T_{\text{total}} = T_{\text{forward}} + T_{\text{backward}} \cdot N_{\text{selected}}$, where $N_{\text{selected}}$ denotes the number of samples used for backpropagation. Thanks to our Sample Efficient Learning Strategy, our method significantly reduces $N_{\text{selected}}$ compared to baselines. For instance, as shown in Table B, our method requires up to 45% fewer backpropagation than EATA, resulting in lower overall computation time.
> *  **Resource Requirements.** As shown in Table B, we adopt LoRA for parameter updates, thereby limiting the trainable parameters to a small fraction of the full model. This design keeps memory usage and computational overhead low, especially when compared to full-parameter tuning. Consequently, our method enables practical adaptation in resource-constrained environments.
>
> Table B. Experimental results of our proposed method in the **Online setting (batch size=1)**.
> | Methods|Average R-Lsum|Trainable parameters |#Backwards|
> |-|-|-|-|
> |Llama3.1-8B|0.1720|-|-|
> |Tent|0.1834|3.41M|5000|
> |EATA|0.1809|3.41M|4634|
> |Ours| **0.1917**|3.41M|**2541**|
>
> ****
> We sincerely hope our clarifications above have addressed your concerns. We would be grateful if you could kindly reconsider the evaluation of our paper.

---

> > ### Comment · Reviewer_CEYm · 2025-04-04
> >
> > Thank you for the detailed and thoughtful responses. The additional comparison with COME, analyses of time complexity and resource usage, and demonstration of performance in an online streaming environment have addressed most of my major concerns. However, some issues have remained, which are similar to those raised by Reviewer BSk7:
> > * Lack of theoretical formulation for PPL minimization: While the effectiveness of PPL minimization presented in the paper may appear empirically plausible, it lacks a rigorous theoretical formulation, which undermines its persuasiveness.
> > * Insufficient distinction between the proposed TTL and TTA: The significant difference between the proposed TTL and TTA remains unclear.
> > * Need for experiments on tasks involving longer responses and i.i.d. conditions: Additional results are required for tasks with extended sequences and an i.i.d. setup.
> >
> > Based on these points, I will maintain my current rating, but will positively reconsider upon seeing these concerns resolved.

---

> > > ### Author Response · Authors · 2025-04-05
> > >
> > > Dear Reviewer,
> > >
> > > Thank you very much for your prompt response to our rebuttal. We hope to make the following further responses to your concerns and sincerely hope you would be satisfied.
> > >
> > > >Q1. Lack of theoretical formulation for PPL minimization.
> > >
> > > **A1.** Thanks for you suggestion. We provide additional theoretical analysis to justify the connection between input optimization and output PPL reduction.
> > > * **Autoregressive Training Dynamics.** The standard next-token prediction objective makes model predictions inherently conditional on previous context quality. Thus, our TTL seeks to optimize the model by input PPL minimization, which will boost output PPL with improved context predition.  In other words, an improved context representation of input (also the context to next token) will benefit more accurate next-token predictions. As a result, the TTL objective is also expected to reduce output PPL.
> > > * **Gradient-Based Theoretical Analysis.** We formalize the intuition that question-conditioned updates benefit answer predictions under a key assumption. Let $\theta' = \theta - \eta \nabla_\theta (-\log P(q;\theta))$ denote the updated parameters after a single TTL step, where $q$ is the question. Using a first-order Taylor expansion:
> > > $\log P_{\theta'}(a|q) \approx \log P_\theta(a|q) + \eta \underbrace{\left[ \nabla_\theta \log P(q;\theta) \right]^\top \nabla_\theta \log P_\theta(a|q)}_{\text{Cross-gradient term}} + \mathcal{O}(\eta^2)$, where $a$ is the answer to the qestion $q$. Our core assumption is that
> > >
> > > $\langle \nabla_q, \nabla_a \rangle=\left[ \nabla_\theta \log P(q;\theta) \right]^\top \nabla_\theta \log P_\theta(a|q) \geq 0$ for question-answer pairs with strong semantic alignment. Under this condition, the cross-gradient term becomes non-negative, guaranteeing: $\log P_{\theta'}(a|q) \geq \log P_\theta(a|q)$ for small $\eta$.
> > > * **Empirical Validation of Assumption:** We compute this gradient inner product using 100 batches QA-pairs from the Geography over Llama3.1-8B. Results show 92% of batch-samples satisfy the non-negativity condition, with average $\langle \nabla_q, \nabla_a \rangle = +23.36$. This strongly supports our theoretical premise.
> > >
> > > We will include this analysis in the revised manuscript, with expanded derivations and statistical details.
> > >
> > > >Q2. Insufficient distinction between the proposed TTL and TTA.
> > >
> > > **A2.** We respectfully further clarify the key distinctions between TTL and TTA:
> > > * TTA focuses on unsupervised adaptation at the output level. This, however, easily suffers from error accumulation without reliable supervision and results in performance degradation. In contrast, **we leverage the auto-regressive nature of language processing tasks, and design a supervised objective for LLMs based on the model inputs to stably guide the model adaptation during testing**.
> > > * Our empirical evaluations in Table 2 of the main paper also demonstrate that, while TTA methods improve model performance in certain domains, they degrade the performance in many others. **Similar results are observed in our ablation study from Tab. A.** In contrast, our TTL scheme demonstrates superior stability, and consistently improves performance across domains and across various models. We will make this clearer in the revised paper.
> > >
> > > Tab. A: Comparison between entropy and PPL minimization on DomainBench.
> > > | Methods|Geo.|Agri.|Med.|Fin.|
> > > |-|-|-|-|-|
> > > |Llama3.1-8B|0.2450|0.0834|0.1265|0.2329|
> > > |Entropy|0.0778|0.0067|0.0105|0.0372|
> > > |PPL|**0.3190**|**0.1255**|**0.2326**|**0.3222**|
> > >
> > > >Q3. Need for experiments on tasks involving longer responses and i.i.d. conditions.
> > >
> > > **A3.** Here are the clarifications:
> > > * **Our experiments already follow the i.i.d assumption, per your suggestion, and each test sample is drawn *Independently from an Identical Distribution***. Our setup is adopted from existing TTA methods, where the model adapts continuously to the target domain, allowing the model to capture domain-invariant features by learning from more test samples.
> > > * We provide more challenging evaluations when the model is adapted and tested on a different domain in Tab. B. **The results underscore the effectiveness of TTL to learn domain-generalizable features**. We will make this clearer in the revision.
> > > * As suggested, **we conducte experiments on longer response generation using LongWriter [r1]. Our method also shows a notable improvement on Llama3.1-8B (0.1637→0.1709)**. We will include an analysis of the longer response generation experiments in the revised manuscript.
> > >
> > > Tab. B: Results on DomainBench.
> > > | Methods|Geo.|Agri.|Med.|Fin.|
> > > |-|-|-|-|-|
> > > |Llama3.1-8B|0.2450|0.0834|0.1265|0.2329|
> > > |Ours (Train on the Geo./Agri./Med./Fin.)|0.3212|0.1319|0.2372|0.3242|
> > > |Ours (Train on Geo.)|0.3212|0.1227|0.1740|0.3055|
> > >
> > > [r1] LongWriter: Unleashing 10,000+ Word Generation from Long Context LLMs, arxiv 2024.
> > >
> > > We sincerely hope our clarifications above have addressed your concerns.
> > >
> > > Best,
> > >
> > > Authors

---

### Official Review · Reviewer_BSk7 · 2025-03-11

**Overall Recommendation:** 2

**Summary:**

This paper proposes a test-time learning scheme for LLMs that minimizes input perplexity (instead of entropy in Test-Time Adaptation) for unlabeled data at test time for better domain adaptation. In addition, the authors draw an insight showing that high-perplexity inputs are more informative for optimization and thus propose to select those for test-time updates. Extensive empirical experiments suggest that the proposed method is generally effective.

**Claims And Evidence:**

In Sec. 4.1, the authors claimed "This improved representation facilitates more accurate and confident next-token predictions, thereby reducing conditional output perplexity", which is not formally derived or supported. After the minimization of input perplexity, the model parameter $\Theta$ has changed, thus it is different from the real conditional output perplexity $\mathcal{P}(y | x; \Theta)$. So, the claim is not properly justified either theoretically or empirically.

**Essential References Not Discussed:**

Yes. The key contribution is a method that minimizes input perplexity for test time training/adaptation, but there was prior work [R Sennrich'EACL 2012] that considered perplexity minimization for domain adaptation. This could harm the novelty of the paper.

[R Sennrich'EACL 2012] "Perplexity Minimization for Translation Model Domain Adaptation in Statistical Machine Translation"

**Experimental Designs Or Analyses:**

Yes, I have checked Section 5 Experiments. The setting of model training or restoration is not specified well. For example, in a typical setting of Test-Time Training, the model will be reverted to its original weight after training of each test sample. However, this setting of model optimization is not described in the paper.

**Methods And Evaluation Criteria:**

Rouge scores and accuracy using extract match make sense. But, since the paper uses perplexity minimization for optimization, the perplexity itself can be one meaningful metric which is not presented in the paper.

**Other Comments Or Suggestions:**

- Line 161: the model “is” optimized to .. "is" should be added.

**Other Strengths And Weaknesses:**

## Other Weaknesses

- The justification of batch setting is not discussed, as in real-world scenarios testing data often comes in a streaming fashion.
- Although the key claim of this paper is to minimize perplexity for test input instead of minimizing entropy as prior works, the perplexity is just a monotonic transformation of entropy itself, i.e., $Perplexity(P) = e^{H(P)}$. Thus, minimizing cross entropy directly minimizes perplexity. It is not well-understood what the real difference is and how perplexity benefits the test-time adaptation.

**Questions For Authors:**

- What is the essential difference between TTA and TTL? Minimizing perplexity or entropy is almost the same.
- Can you show a comparison of Figure 1b, which presents an updated input and output perplexity after input perplexity minimization? This is helpful for understanding the effectiveness of the correlation after TTL.
- What does Figure 1c mean? Are low perplexity samples and high perplexity samples mixed together in the training or separate?

**Relation To Broader Scientific Literature:**

The finding that minimizing perplexity is performing better than minimizing cross-entropy is relevant to the test-time adaptation literature and test-time training literature.

**Theoretical Claims:**

There are no proofs or formal theoretical claims in this paper.

---

> ### Author Rebuttal · Authors · 2025-04-01
>
> >Q1. The claim "This improved representation ..., thereby reducing conditional output PPL" is not properly justified either theoretically or empirically.
>
> **A1.** We thank the reviewer for this important observation. We agree that the original statement regarding conditional output PPL reduction requires more precise formulation and supporting evidence. Please allow us to clarify:
> * **Causal Relationship Clarification.** We have revised the phrase "thereby reducing conditional output perplexity" to **"This improved representation facilitates more accurate and confident next-token predictions, which is expected to reduce output perplexity".**
> * **Theoretical Motivation Clarification.** In our study, we observe a strong positive correlation between input and output PPL (see Fig.1b). Relying on this, we seek to reduce the output PPL by input PPL minimization.
>
> * **Empirical Validation.** We compare changes in both input and output PPL between our method and the original LLM in Tab. A. **Our method reduces input PPL and consistently decreases output PPL.**
>
> Tab. A: PPL results on DomainBench.
> || Methods|Geo.|Agri.|Med.|Fin.|
> |-|-|-|-|-|-|
> |Input PPL|Llama3.1-8B|187.3|933.2|43.6|323.0|
> ||Ours|6.6|3.9|2.0|5.5|
> |Output PPL|Llama3.1-8B|3242.9|205247.1|7.7|1208976.9|
> ||Ours| 2483.9|156232.3|6.6|242617.0|
>
> >Q2. Add PPL as metric.
>
> **A2.** As suggested, we include PPL as a metric in Tab. B. **Our method outperforms existing methods in terms of PPL**.
>
> Tab. B: PPL results on DomainBench.
> | Methods|Geo.|Agri.|Med.|Fin.|
> |-|-|-|-|-|
> |Llama3.1-8B|3242.9|205247.1|7.7|1208976.9|
> |EATA|4692.1|628196.9|2937.1|11444422.0|
> |Ours|**2483.9**|**156232.3**|**6.6**|**242617.0**|
>
> >Q3. The training/restoration setting is not clear.
>
> **A3.** In our method, the model weights are not restored to their original state after processing each test sample. Specifically,
> * **Offline Setting.** All test data are processed at once, and the model is updated using all available test samples before any testing.
> * **Online Setting.** Test samples are processed sequentially, with the model updated after each test sample or mini-batch, and the model parameters are never reset.
>
> >Q4. Differences of the proposed method from [R Sennrich].
>
> **A4.** Both works use PPL minimization, and our method differs from [R Sennrich] as:
> * **Problem Setting.** We focus on **adapting LLMs at test time using unlabeled test data** to handle distribution shifts. In contrast, [R Sennrich] assumes access to **labeled source-domain training data** and optimizes translation models from a static source corpus.
> * **Optimization Objectives.** We adopt **input PPL minimization** as the optimization objective, enabling efficient self-adaptation of LLMs to target domains during testing. Instead, [R Sennrich] optimizes **output PPL** using labeled data.
>
> >Q5. Justification of batch setting needed.
>
> **A5.** We evaluate batch size in the **online setting** (bs=1 vs 100) under streaming-like scenarios. **With bs=1, the model updates after each test sample. Results demonstrate increased update steps（1514→2541), while our TTL remains effective (0.2040→0.1917).**
>
> >Q6&Q7. Differences between TTA and TTL? Also, PPL and entropy?
>
> **A6&A7.** Both TTA and TTL adapt models without labeled data, but they differ in key aspects:
> * **Objective Function.** Most TTA methods minimize the output entropy $H(P(y|x))$. TTL minimize the PPL of the input sequence $\mathcal{P}(x)=e^{1/T\sum_{t=1}^TH_{CE}^{(t)}}$, where $H_{CE}^{(t)}$ is the token-level cross-entropy.
> * **Task-Specific Adaptation.** TTA seeks to improve output confidence but cannot optimize the input representation quality that is essential for LLMs. In contrast, TTL minimizes input PPL to refine the internal representation of the input sequence to encourage more accurate predictions.
>
> >Q8. Can you show a comparison of Fig.1b, which presents an updated input and output PPL after input PPL minimization?
>
> **A8.**  Fig. 1(b) shows **a strong positive correlation between input and output PPL** on DomainBench using Llama3.1-8B with varying degrees of training. We compare the changes in both input and output PPL of our method with the original LLM in Tab. A, **our method reduces input PPL and consistently decreases output PPL**.
>
> >Q9. Fig.1c meanings and Training Protocol.
>
> **A9.** Clarifications are below:
> * Fig. 1c shows how the selection of different proportions of test samples (X-axis: top/bottom p% by PPL) would impact the final performance  for TTL  (Y-axis: evaluated on the full test set).
> * **Training Protocol.** We pre-sort test samples by PPL, select a given proportion, **randomly shuffle the subset**, and perform TTL. The adapted model is then evaluated on all test data.
> ****
> We sincerely hope our clarifications above have addressed your concerns. We would be grateful if you could kindly reconsider the evaluation of our paper.

---

> > ### Comment · Reviewer_BSk7 · 2025-04-03
> >
> > Thank the authors for the response, but most of my concerns are not well-addressed. I think my current score is fair.
> >
> > - The clarification does not make the theoretical formulation clearer or convincing. Although empirically it may hold, the conditional output PPL reduction needs a more rigorous formulation to make the claim solid.
> >
> > - I suggest that the authors add PPL results for more challenging tasks that have longer responses like ReasoningBench.
> >
> > - If the model weights are not restored after each test sample, some continuity of the test sequence or the distribution should be assumed. However, practically, the test samples are often i.i.d. which means learning on one test sample (minimizing input perplexity for one specific test sample) does not provide any guarantee that it will benefit other test samples. The model is often restored in almost all test-time training settings.
> > - I still couldn’t see sufficient distinction/novelty from the proposed TTL compared with TTA. I think ablations regarding how entropy and perplexity minimization differ are significant for understanding the effectiveness of this work.

---

> > > ### Author Response · Authors · 2025-04-05
> > >
> > > >Q1. The clarification does not make the theoretical formulation clearer or convincing.
> > >
> > > **A1.** Thanks for you suggestions. We provide additional theoretical analysis to justify the connection between input optimization and output PPL reduction.
> > > * **Autoregressive Training Dynamics.** The standard next-token prediction objective makes model predictions inherently conditional on previous context quality. Thus, our TTL seeks to optimize the model by input PPL minimization, which will boost output PPL with improved context predition.  In other words, an improved context representation of input (also the context to next token) will benefit more accurate next-token predictions. As a result, the TTL objective is also expected to reduce output PPL.
> > > * **Gradient-Based Theoretical Analysis.** We formalize the intuition that question-conditioned updates benefit answer predictions under a key assumption. Let $\theta' = \theta - \eta \nabla_\theta (-\log P(q;\theta))$ denote the updated parameters after a single TTL step, where $q$ is the question. Using a first-order Taylor expansion:
> > > $\log P_{\theta'}(a|q) \approx \log P_\theta(a|q) + \eta \underbrace{\left[ \nabla_\theta \log P(q;\theta) \right]^\top \nabla_\theta \log P_\theta(a|q)}_{\text{Cross-gradient term}} + \mathcal{O}(\eta^2)$, where $a$ is the answer to the qestion $q$. Our core assumption is that
> > >
> > > $\langle \nabla_q, \nabla_a \rangle=\left[ \nabla_\theta \log P(q;\theta) \right]^\top \nabla_\theta \log P_\theta(a|q) \geq 0$ for question-answer pairs with strong semantic alignment. Under this condition, the cross-gradient term becomes non-negative, guaranteeing: $\log P_{\theta'}(a|q) \geq \log P_\theta(a|q)$ for small $\eta$.
> > > * **Empirical Validation of Assumption:** We compute this gradient inner product using 100 batches QA-pairs from the Geography over Llama3.1-8B. Results show 92% of batch-samples satisfy the non-negativity condition, with average $\langle \nabla_q, \nabla_a \rangle = +23.36$. This strongly supports our theoretical premise.
> > >
> > > We will include this analysis in the revised manuscript, with expanded derivations and statistical details.
> > >
> > > >Q2. Need to add PPL results for the tasks that have longer responses like ReasoningBench.
> > >
> > > **A2.** As suggested, we further include the results on ReasoningBench in terms of PPL in Tab. A. **Our method still outperforms existing methods in terms of PPL**.
> > >
> > > Tab. A: PPL results on ReasoningBench.
> > > | Methods|GSM8K|MetaMath|Logiqa|
> > > |-|-|-|-|
> > > |Llama3.1-8B|4.3|2.2|20.2|
> > > |Tent|5.5|2.8|204092.9|
> > > |EATA|8.6|409.8|5643358262027.4|
> > > |Ours|**4.0**|**2.1**|**10.9**|
> > >
> > > >Q3. The test samples are often i.i.d. which means learning on one test sample does not provide any guarantee that it will benefit other test samples.
> > >
> > > **A3.** Here are the clarifications:
> > > * **Our experiments already follow the i.i.d assumption, per your suggestion, and each test sample is drawn *Independently from an Identical Distribution***. Our setup is adopted from existing TTA methods, where the model adapts continuously to the target domain, allowing the model to capture domain-invariant features by learning from more test samples.
> > > * We agree that continuous learning may encounter issues like overfitting, but frequent restoration may also otherwise suffer from underfitting. We believe that each setting has its considerations and concerns, and the debate is out of the scope of our paper.
> > > * We provide more challenging evaluations when the model is adapted and tested on a different domain in Tab. B. **The results show the effectiveness of TTL to learn domain-generalizable features**. We will make this clearer in the revision.
> > >
> > > Tab. B: Results on DomainBench.
> > > | Methods|Geo.|Agri.|Med.|Fin.|
> > > |-|-|-|-|-|
> > > |Llama3.1-8B|0.2450|0.0834|0.1265|0.2329|
> > > |Ours (Train on the Geo./Agri./Med./Fin.)|0.3212|0.1319|0.2372|0.3242|
> > > |Ours (Train on Geo.)|0.3212|0.1227|0.1740|0.3055|
> > >
> > > >Q4. Need to discuss distinction/novelty from TTL compared with TTA, add ablations about entropy and perplexity minimization.
> > >
> > > **A4.** We further clarify the key distinctions between TTL and TTA:
> > > * TTA focuses on **unsupervised** adaptation at the output level. This, however, easily suffers from error accumulation without reliable supervision and results in performance degradation. In contrast, we use the auto-regressive nature of language processing tasks and design a **supervised** objective for LLMs based on inputs to stably guide the adaptation during testing.
> > > * TTA methods improve performance in certain domains but **degrade** the performance in many others (see Tab. 2). **Similar results are observed in ablation study from Tab. C.** In contrast, TTL shows superior stability and **consistently improves** performance across domains and across various models.
> > >
> > > Tab. C: Entropy and PPL minimization on DomainBench.
> > > | Methods|Geo.|Agri.|Med.|Fin.|
> > > |-|-|-|-|-|
> > > |Llama3.1-8B|0.2450|0.0834|0.1265|0.2329|
> > > |Entropy|0.0778|0.0067|0.0105|0.0372|
> > > |PPL|**0.3190**|**0.1255**|**0.2326**|**0.3222**|

---

### Decision · Program_Chairs · 2025-05-01

**Decision:**

Accept (poster)

**Comment:**

This paper introduces Test-Time Learning (TTL) for large language models (LLMs), a method that adapts LLMs to domain shifts using only unlabeled test data. The key idea is to minimize input perplexity (PPL) rather than output entropy, as a self-supervised objective, leveraging the observation that high-PPL inputs are more informative for adaptation. The authors propose a sample-efficient strategy that prioritizes such inputs and employs LoRA for stable updates, avoiding catastrophic forgetting. They also introduce AdaptEval, a benchmark covering domain-specific, instruction-following, and reasoning tasks. Experiments show consistent improvements over baselines, demonstrating the method’s effectiveness in real-world scenarios where labeled data is scarce.

Reviewers acknowledged the practical contributions but raised concerns: (1) Theoretical gaps in linking input PPL minimization to improved output predictions; (2) Unclear novelty compared to prior TTA and perplexity-based methods; (3) Limited comparisons with recent TTA techniques (e.g., COME) and alternative approaches like fine-tuning/RAG;

After the rebuttal, the authors addressed concerns by: (1) Adding gradient-based theoretical analysis to justify the input-output PPL connection, supported by empirical validation; (2) Expanding experiments to include comparisons with COME (ICLR’25) and evaluations on longer-response tasks; (3) Clarifying distinctions between TTL and TTA. However, some concerns remain. The theoretical claims still lack rigor, the novelty of PPL minimization versus entropy minimization needs deeper ablation, and the comparison with TTA methods needs to be included.

Most of the reviewers agreed that some key concerns have been addressed, while the presentation needs substantial modification at current stage. Given these considerations, I recommend weak acceptance. The authors should improve the paper by incorporating all the discussions and feedback during the rebuttal.